# Structural basis for energy transduction by respiratory alternative complex III

Joana S. Sousa[1], Filipa Calisto[2], Julian D. Langer[3,4], Deryck J. Mills[1], Patrícia N. Refojo[2], Miguel Teixeira [2], Werner Kühlbrandt[1], Janet Vonck [1] & Manuela M. Pereira [2,5]

Electron transfer in respiratory chains generates the electrochemical potential that serves as energy source for the cell. Prokaryotes can use a wide range of electron donors and acceptors and may have alternative complexes performing the same catalytic reactions as the mitochondrial complexes. This is the case for the alternative complex III (ACIII), a quinol:cytochrome *c*/HiPIP oxidoreductase. In order to understand the catalytic mechanism of this respiratory enzyme, we determined the structure of ACIII from *Rhodothermus marinus* at 3.9 Å resolution by single-particle cryo-electron microscopy. ACIII presents a so-far unique structure, for which we establish the arrangement of the cofactors (four iron–sulfur clusters and six *c*-type hemes) and propose the location of the quinol-binding site and the presence of two putative proton pathways in the membrane. Altogether, this structure provides insights into a mechanism for energy transduction and introduces ACIII as a redox-driven proton pump.

[1] Department of Structural Biology, Max Planck Institute of Biophysics, Max-von-Laue Str. 3, 60438 Frankfurt am Main, Germany. [2] Instituto de Tecnologia Química e Biológica–António Xavier, Universidade Nova de Lisboa, ITQB NOVA, Av. da República EAN, 2780-157 Oeiras, Portugal. [3] Department of Molecular Membrane Biology, Max Planck Institute of Biophysics, Max-von-Laue Str. 3, 60438 Frankfurt am Main, Germany. [4] Proteomics, Max Planck Institute for Brain Research, Max-von-Laue Str. 4, 60438 Frankfurt am Main, Germany. [5] Departamento de Química e Bioquímica, Faculdade de Ciências, Universidade de Lisboa, 1749-016 Lisboa, Portugal. These authors contributed equally: Joana S. Sousa, Filipa Calisto. Correspondence and requests for materials should be addressed to J.V. (email: janet.vonck@biophys.mpg.de) or to M.M.P. (email: mpereira@itqb.unl.pt)

Energy is at the basis of life as all organisms depend on constant energy transduction mechanisms to grow and reproduce themselves. Electron transfer respiratory chains generate the transmembrane difference of electrochemical potential that is the energy source for ATP synthesis, solute transport, and motility. In eukaryotes, the respiratory chain is located in mitochondria and is classically viewed as a linear composition of four electron transfer complexes, known as complex I–IV. However, in many organisms respiratory chains differ in composition and organization[1]. Prokaryotes can use a wide range of electron donors and acceptors and may have alternative complexes performing essentially the same catalytic reactions as the canonical mitochondrial complexes[2]. The diversity and apparent redundancy of prokaryotic respiratory chains reflects the versatility and robustness of their organisms.

The oxidation of quinol with subsequent reduction of cytochrome $c$ seemed until 1999 to evade the paradigm of diversified respiratory chains in prokaryotes. The reaction was thought to be exclusively catalyzed by the $bc_1/b_6f$ complex[2], also known as complex III. This notion changed with the identification of alternative complex III (ACIII), a quinol:cytochrome $c$/high potential iron–sulfur protein (HiPIP) oxidoreductase that was identified in *Rhodothermus marinus*[3]. ACIII is widespread in bacteria[3–7] and mainly present in organisms that lack the $bc_1/b_6f$ complex. Nonetheless, the genes coding for ACIII and complex III coexist in some species[1,8]. Their expression in these may depend on the cellular metabolic needs, as it has been reported for respiratory enzymes performing the same catalytic activity, such as complex I and NDH-2 in *Escherichia coli*[9]. In addition to reducing periplasmatic proteins, such as HiPIP[10] and soluble cytochrome $c$[11], ACIII can directly transfer electrons from quinol to the $caa_3$ terminal oxidase without the intervention of any soluble electron carrier[8]. Besides the functional linkage of ACIII with the $caa_3$ oxygen reductase, the structural association of the two complexes into a supercomplex has also been proposed in *R. marinus* membranes based on biochemical and functional experiments[8]. These observations agree with the fact that gene clusters encoding ACIII are frequently followed by gene clusters encoding oxygen reductases[8,12].

Even though ACIII is functionally equivalent to the cytochrome $bc_1$ complex[3,4,13,14], these two enzymes are structurally unrelated. In *R. marinus*, ACIII is composed of seven subunits encoded by the *Act* gene cluster[8], six of which are conserved across species (ActA to ActF). ActA and ActE are cytochromes with five and one $c$-type hemes, respectively. The largest subunit in the complex is ActB, which is composed of two domains. These domains, designated B1 and B2, are homologous to the catalytic subunit and to the iron–sulfur protein of the members of the complex iron–sulfur molybdoenzyme (CISM) family[14], respectively. ActC and ActF are membrane subunits and are homologous to subunit C of the polysulfide reductase protein (PsrC), also a member of the CISM family[7,14]. Finally, ActD and ActG are predicted to be transmembrane proteins, without redox cofactors, and seem to be present exclusively in ACIII[15]. ACIII is composed by a combination of proteins present in other respiratory complexes, which reinforces the modularity concept of energy transducing machines[12]. However, no structural characterization of this family of respiratory enzymes was performed to date.

In this work, we describe a 3.9 Å cryo-EM structure of ACIII from *R. marinus*. Our structure shows that ACIII meets all requirements for an energy-transducing machine that couples quinol oxidation to translocation of protons across the membrane.

## Results

**Overall structure of ACIII**. ACIII was purified from *R. marinus* membranes solubilized with *n*-dodecyl β-D-maltoside (DDM) (Supplementary Fig. 1a) and studied by cryo-EM. Grids showed ACIII as compact particles with a good distribution in the ice in random orientations (Supplementary Fig. 1b, d). By 2D and 3D classification routines, two populations of particles were identified in the data set (Supplementary Fig. 1c). The largest set, with almost 40% of the particles, produced a map at an average resolution of 3.9 Å (Supplementary Fig. 1e, f and Supplementary Fig. 2a). At this resolution, α-helices, β-sheets, side chains for most residues, and the iron–sulfur clusters and heme groups and their protein ligands are easy to recognize (Supplementary Fig. 2b-e). The only exception are glutamate and aspartate side chains, which are highly radiation sensitive[16–18]. This map was used to solve the ACIII structure (Fig. 1 and Supplementary Table 1). The smaller set of particles (7% of the initial data set) produced a map at 20 Å resolution that displays a large extra density and is compatible with an ACIII–$caa_3$ supercomplex that was previously proposed to exist[8] (Fig. 2).

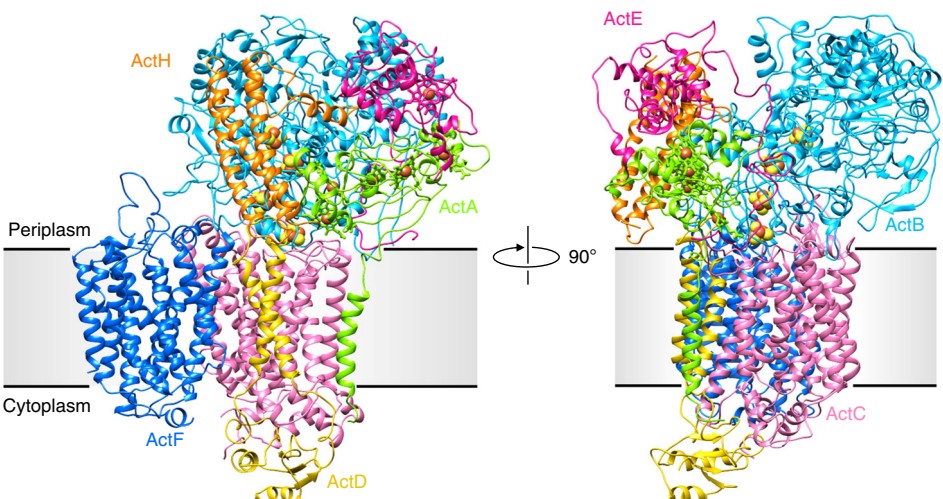

**Fig. 1** Overall structure of ACIII. ACIII is composed of at least seven subunits; the peripheral subunits ActA (green), ActB (cyan), ActE (magenta), and ActH (orange), and the membrane subunits ActC (pink), ActD (yellow), and ActF (blue). ActB coordinates four iron–sulfur clusters (orange/yellow spheres). ActA binds five hemes (green sticks and orange spheres) and ActE one heme (magenta sticks and orange sphere). The gray band indicates the membrane

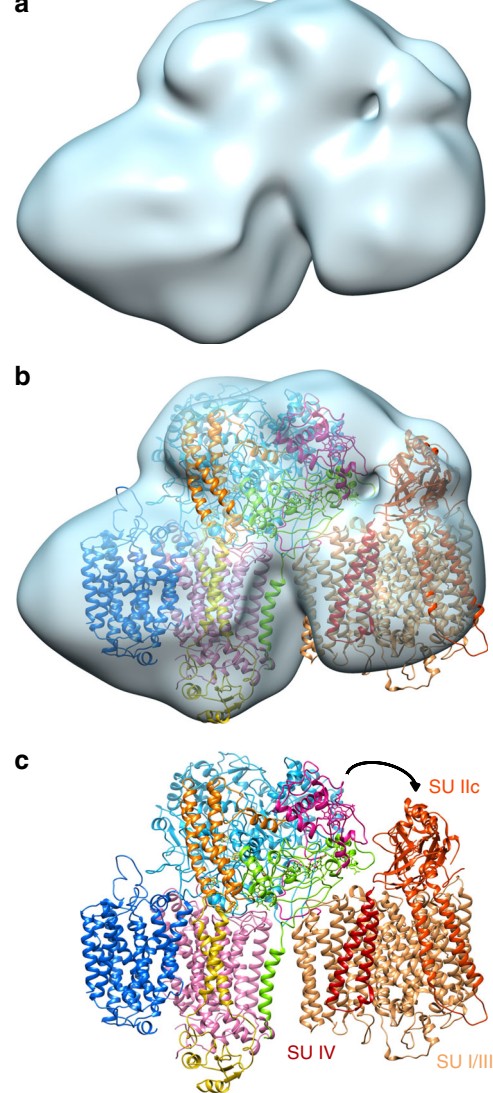

**Fig. 2** ACIII–$caa_3$ supercomplex. **a** Cryo-EM map of ACIII–$caa_3$ supercomplex at 20 Å resolution. **b** Fit of ACIII model (this work) and *Thermus thermophilus* $caa_3$ oxygen reductase (PDB 2YEV)[32] to the ACIII–$caa_3$ supercomplex map. **c** Fitted models without cryo-EM map. The relative position of ActE and subunit IIc from the $caa_3$ oxygen reductase in the supercomplex is favorable for direct electron transfer between the two complexes (black arrow)

ACIII is an L-shaped complex of ~130 Å by ~110 Å (Supplementary Fig. 2a) with a membrane domain and two peripheral domains facing opposite sides of the membrane. The six subunits ActABCDEF[12,14], encoded by the *Act* gene cluster, were traced in the map. Subunits ActA, ActB, and ActE in the large peripheral domain accommodate all the prosthetic groups in the complex, for a total of six hemes and four iron–sulfur clusters. Since ActA and ActE are *c*-type cytochromes, the large peripheral domain is facing the periplasm, a conclusion also supported by the topological prediction of the membrane subunits. The two homologous subunits, ActC and ActF, with ten transmembrane helices (TMH) each, and the smaller subunit ActD with two TMHs form the major membrane domain. An additional TMH composed of the N-terminal residues of ActA brings the total number of TMHs in the complex to 23. No density was identified for the seventh subunit ActG, although its presence was confirmed by peptide mass fingerprinting (Supplementary Fig. 3).

Notably, as the only subunit whose gene is not present in all *Act* gene clusters[14], this subunit does not appear to be essential for catalysis. Two additional densities are visible in the map. One of these densities was identified by peptide mass fingerprinting as the protein Rmar_1979, annotated as a hypothetical protein in the NCBI Protein database, that we named ActH (Fig. 3). The second additional subunit is located in the periplasm, above ActE (Supplementary Fig. 2a), but due to the lower resolution of this map region an assignment was not possible. We are confident that this unidentified density is not ActG since it is largely exposed to the solvent, while ActG is predicted to span the membrane with one TMH.

The structure shows that ACIII subunits are organized into two functional modules: (i) the electron transfer module, composed of subunits ActA, ActB, ActE, ActH, and one unidentified subunit and (ii) a module for membrane attachment, quinol-binding and ion translocation, comprising membrane subunits ActC, ActD, and ActF.

**Electron transfer module.** ActA is a pentaheme cytochrome *c*, anchored to the membrane by an N-terminal TMH (Supplementary Fig. 4a). The helix appears to be isolated in the membrane, in contact only with loops of subunits ActC and ActD on either side of the membrane. The comparatively weak map density suggests that this helix is flexible (Supplementary Fig. 5).

The soluble domain of ActA harbors five heme groups and is associated with ActB and ActE. Two subdomains are observed from residues 45 to 89 and residues 121 to 165. These subdomains are related by a two-fold rotational axis of pseudo-symmetry and are most likely a result of early gene duplication (Supplementary Fig. 4a).

The ActA hemes are arranged in a linear way, forming a ~45 Å wire. Most hemes are coordinated by two histidines, except heme I, for which one histidine and one methionine serve as axial ligands (Supplementary Fig. 4a). The two pseudo-two-fold symmetric subdomains accommodate hemes II–III and IV–V respectively, which are organized in two di-heme elbow motifs, with a ~6 Å edge-to-edge distance (Fig. 4 and Supplementary Fig. 4a). The porphyrin rings of the central hemes III and IV are parallel, at an edge-to-edge distance <4 Å. The relative position of these two hemes is identical to the architecture observed for the split-Soret cytochrome from *Desulfovibrio desulfuricans* ATC27774[19]. This arrangement is most likely responsible for the split Soret effect observed in the UV–visible absorbance spectrum of ACIII[3]. Heme I is <6 Å away from heme II and does not integrate any typical heme motif (Fig. 4 and Supplementary Fig. 4a). This heme is located at the periphery of subunit ActA, at the interface with the large periplasmic subunit ActB, and is partially occluded by the N-terminus of ActE. The apparent reduction potential of these hemes was previously shown to range from −45 mV to +230 mV, at neutral pH[3].

The largest ACIII subunit, ActB, contains almost 1000 residues. The structure corroborates the previous N-terminal sequencing results[13] that suggested the absence of the first 74 amino acid residues in the mature polypeptide. This N-terminal cleavage removes the twin arginine translocase (Tat) signal peptide[13], which is consistent with the periplasmic localization of the subunit.

ActB is in contact with all other subunits and, as anticipated, consists of two distinct domains. The N-terminal domain B1 resembles molybdopterin-containing proteins and includes residues 75–735. In the C-terminal domain B2, from residues 736 to 1039, one [3Fe–4S] cluster (FeS1) and three [4Fe–4S] clusters (FeS2–4)[13,14] are observed (Fig. 4). As predicted[13], no prosthetic groups are present in domain B1. The four iron–sulfur clusters

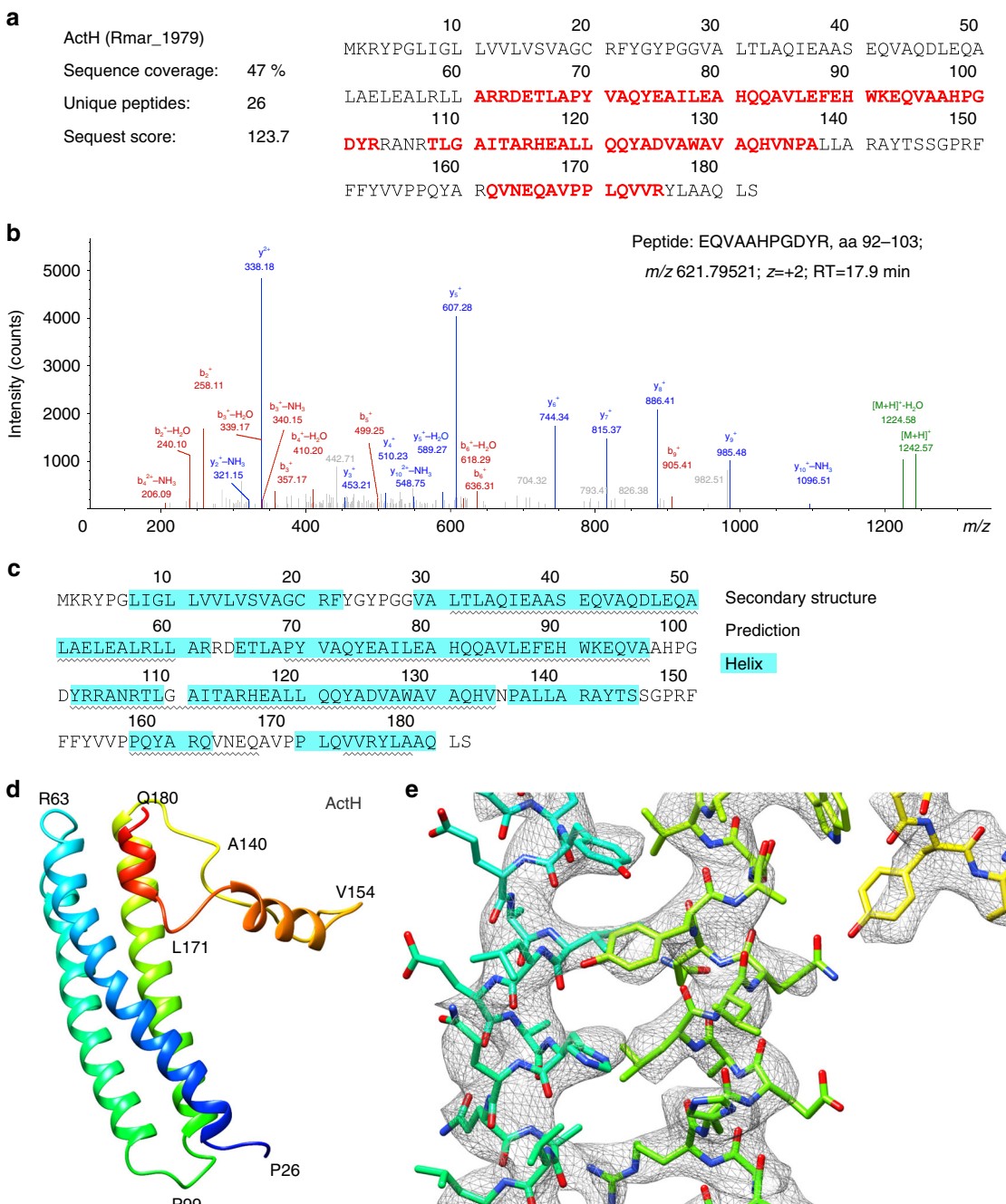

**Fig. 3** Identification of ActH by mass spectrometry. **a** ActH (Rmar_1979, Uniprot identifier: D0MKF0) identification using LC–MS/MS (sequence coverage: 47%, matched peptides highlighted in red). **b** Annotated MS/MS spectrum for peptide EQVAAHPGDYR, corresponding to amino acid residues 92–103 of ActH. **c** Secondary structure predicted by PredictProtein with helices highlighted in cyan. Helices identified in the map are indicated by a wavy line. **d** ActH rainbow-colored from blue (N-terminus) to red (C-terminus). The N-terminal domain is a three-helix bundle, with helices of around 30 residues. The remaining 50 residues form a long loop located in the cleft between ActB, ActA, and ActE and two short helices. **e** Cryo-EM density map of ActH with model. Density is seen for most side chains, with the exception of glutamates and aspartates, which are highly radiation sensitive[16–18]

form a ~35 Å electron transfer wire (Fig. 4), starting with FeS1 close to the membrane surface and ending at FeS4, buried ~7 Å beneath the protein surface. FeS1 is ~11 Å from heme I in ActA and a reduction potential of +140 mV was previously determined by an EPR monitored redox titration[4]. Intriguingly, no signals were detected for [4Fe–4S] centers by EPR spectroscopy[4], and the role of clusters FeS2, FeS3, and FeS4 is still elusive.

ActE is a monoheme cytochrome *c*, located on the top of ActA. Its heme, coordinated by a histidine and a methionine (Supplementary Fig. 4c), adds to the heme wire from ActA, with

a ~9 Å minimum distance to heme V (Fig. 4). The reduction potential of this heme is +165 mV at pH 7.5[20].

The N-terminus of ActE is intertwined with that of ActB and reaches the membrane surface at residue Ile28[E] (Fig. 1 and Supplementary Fig. 2a). Electron density for ActE in the membrane is not visible, preventing further modeling. However, the placement of Ile28[E] close to the membrane supports our earlier identification of a so-called lipobox sequence between residues Leu21[E] and Cys24[E] that led us to suggest ActE as a lipoprotein with a lipid bound to Cys24[E20].

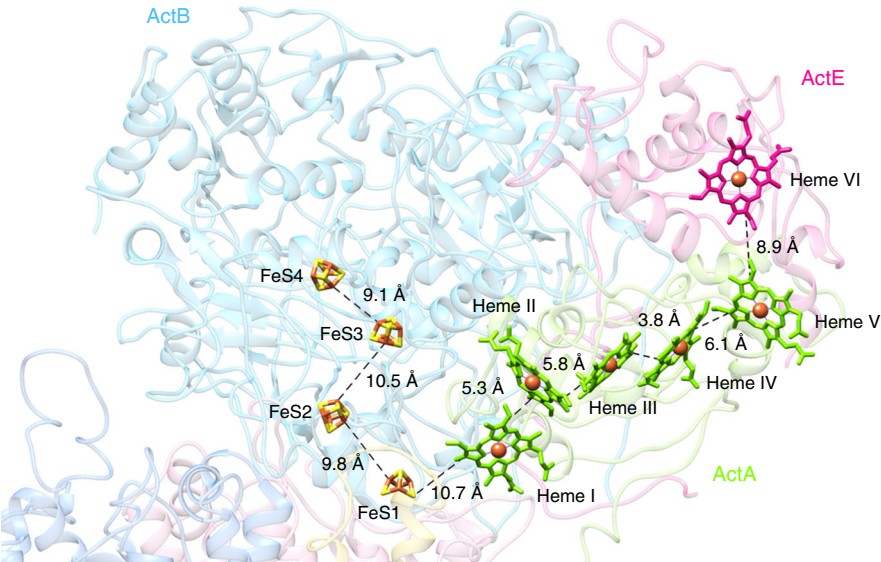

**Fig. 4** Arrangement of prosthetic groups in ACIII. A ~35 Å wire of four iron–sulfur clusters and a ~60 Å six-heme wire form a branched electron transfer chain, with FeS1 receiving electrons directly from the quinol. Broken lines indicate edge-to-edge distances. ActH was removed for clarity. Subunits colored as in Fig. 1

After tracing ActA-F, a prominent density on the periplasmic domain, characterized by a bundle of three long helices was left unassigned. To identify this protein, we made use of combinatorial proteolytic digests combined with liquid chromatography-coupled tandem mass spectrometry. By matching the data to a *R. marinus* database, we identified a total of 152 proteins in the ACIII sample, 62 of which yielded more than two high-scoring, unique peptides. By screening the proteins with a high number of high-scoring peptides, we found the hypothetical protein Rmar_1979 with 26 unique peptides and 47% sequence coverage (Fig. 3a, b). Secondary structure predictions showed three long helices that agreed with the three-helix bundle observed in the structure (Fig. 3c). The good side chain densities in this region enabled us to trace Rmar_1979 in the map (Fig. 3d, e and Supplementary Movie 1), and we thus assigned it as ActH. The protein is completely α-helical and is characterized by a bundle of three long helices interacting with ActA and ActB. Its C-terminal domain contains two more short helices and fills a cleft between ActA, ActB, and ActE. Further bioinformatic analyses (BLASTp) did not reveal any homologs in other prokaryotes, indicating a specific role in *R. marinus* and its close relatives. ActH interacts with all other periplasmic subunits, possibly enhancing complex stability.

**Quinol-binding and ion translocation module**. The two largest membrane subunits, ActC and ActF, have the same fold. They are related by a ~180° rotation around an axis perpendicular to the membrane, forming a dimer with pseudo-two-fold symmetry (Supplementary Fig. 6). Both termini are located in the cytoplasm and contribute to the small cytoplasmic domain of ACIII that is observed in the structure.

ActC and ActF have ten TMHs each, organized in two four-helix bundles and one helix dimer. The four-helix bundles are composed of TMHs 2–5 and TMHs 6–9, which form two structural repeats arranged in a parallel, face-to-face manner (Supplementary Fig. 6a). An amphipathic helix (helix 7a) is observed in the second four-helix bundle of both subunits, between TMHs 7 and 8. The similarity of this fold with that of the polysulfide reductase membrane subunit (PsrC) from *Thermus thermophilus*[21] is noteworthy (Supplementary Fig. 7). The two additional helices (TMHs 1 and 10), which are not present in

PsrC, cross each other at an angle of ~45° at the periphery of the ActC–ActF dimer.

TMHs 2, 5, 6, and 7 of both subunits establish the dimerization interface between ActC and ActF. Furthermore, several elongated densities are observed in the map between the two membrane subunits. These densities are compatible with the presence of lipid (or detergent) molecules, suggesting that the dimerization of ActC and ActF is partially mediated by lipid–protein interactions. In addition to the interactions with ActF, ActC has a large contact surface with ActB and ActD. In contrast, ActF occupies a peripheral position in the structure, with minor contacts with ActB and ActD (Fig. 1).

The putative quinol-binding site is located at the periplasmic side of the membrane, in the first four-helix bundle of ActC, at a distance of 10–12 Å from cluster FeS1 in ActB (Fig. 5, Supplementary Fig. 8 and Supplementary Movie 2). The pocket is mostly hydrophobic and formed by several conserved residues that include Trp82$^C$, Ile86$^C$, Phe89$^C$, Leu166$^C$, Pro136$^C$, and Val250$^C$. Additionally, three highly conserved charged residues are present in this region: His139$^C$, Asp169$^C$, and Asp253$^C$ (Supplementary Fig. 9). The presence of the aspartate residues agrees with the prediction of a quinol-binding site in this region, as acidic residues are required for deprotonation of the quinol[22–24]. Moreover, coordination of the quinol by an essential histidine residue has been shown to be common to many quinone-binding complexes[25]. The quinol pocket is accessible from the membrane by a narrow entry channel delimited by TMH 3 and 4 and the loop connecting them (Supplementary Fig. 8 and Supplementary Movie 2). The location of the quinol pocket also agrees with the quinol-binding site of the homologous PsrC (Supplementary Fig. 7), identified by co-crystallization experiments[21], with a C-alpha RMSD at the periplasmic side of the first four-helix bundle of 1.446 Å. Heme I is more than 20 Å away from the putative quinol-binding pocket, suggesting that FeS1 is the primary electron acceptor from the quinol.

No cofactors are found in the vicinity of ActF. In agreement with this, residues defining the quinol-binding site as in ActC are absent in this subunit (Supplementary Fig. 10 and Supplementary Fig. 11). Interestingly, the loop between TMHs 3 and 4 in ActF shows a large deviation from the conserved fold of ActC and PsrC (Supplementary Fig. 6b) and no pocket or entry channel is

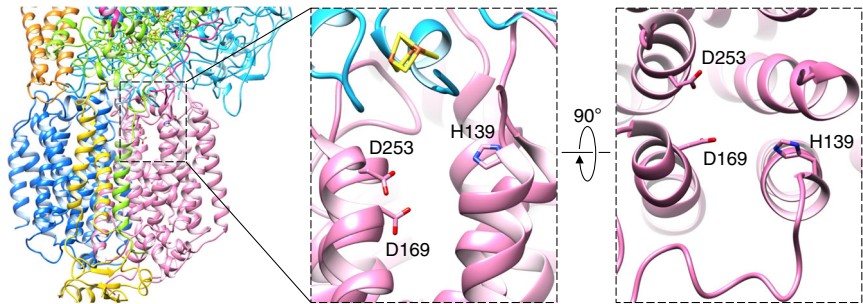

**Fig. 5** Quinol-binding site. Putative quinol-binding site on the periplasmic side of ActC (left), with zoomed views of conserved residues involved in quinol coordination, seen from the membrane (center) and from the periplasm (right). Subunits colored as in Fig. 1. Note that density for aspartate residues is not visible and their side chains have been modeled

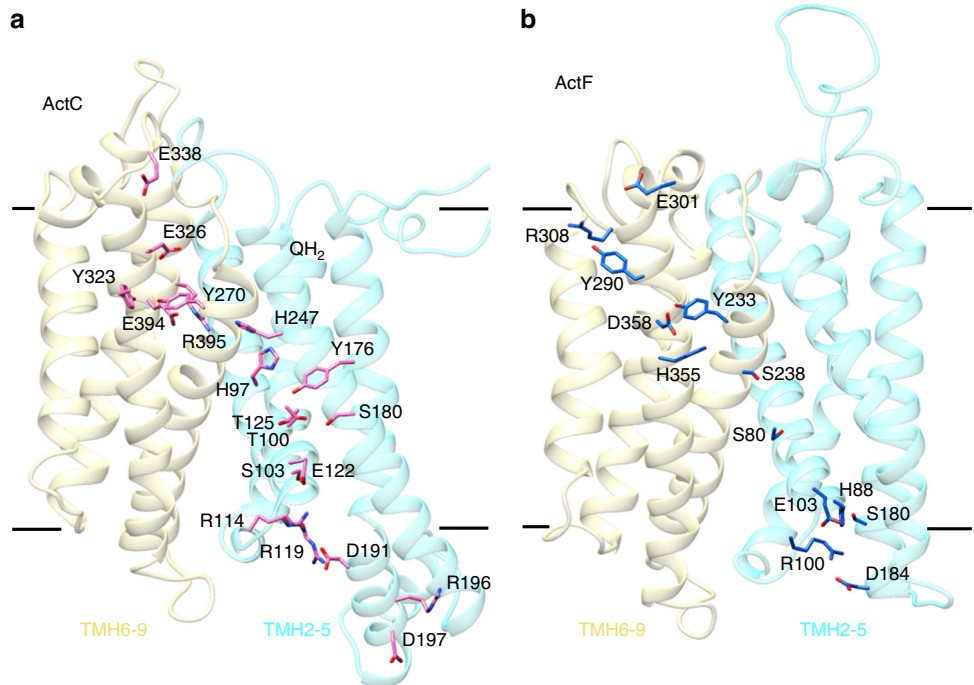

**Fig. 6** Proton pathways. **a** ActC proton-conducting half-channels. The cytoplasmic half-channel (TMHs 2–5) consists of residues Asp197, Arg196, Asp191, Arg119, Glu450, Arg114, Glu122, Ser103, Thr100, Thr125, Ser180, Tyr176, His97, and His247. The periplasmic half-channel (TMHs 6–9) includes residues Arg395, Glu394, Tyr270, Tyr323, Glu326, and Glu338. The approximate position of the quinol pocket is indicated (QH₂). **b** ActF proton-conducting half-channels. Cytoplasmic (TMHs 2–5) and periplasmic (TMHs 6–9) proton-conducting half-channels of ActF include residues Asp184, Arg100, Ser180, Glu103, His88, Ser80, Ser238, His355, Asp358, Tyr233, Tyr290, Arg308, and Glu301. The four-helix bundles are shown in light blue (TMHs 2–5) and yellow (TMHs 6–9). Note that density for aspartate and glutamate residues is not visible and their side chains have been modeled

observed. To confirm the presence of a single binding site for quinol in ACIII, we performed isothermal titration calorimetry (ITC) experiments. ACIII was titrated with 2,3-dimethyl-1,4-naphthoquinone (DMN), a soluble analog of quinone[26]. Analysis of the ITC data (Supplementary Fig. 12) revealed an ideal fitting by a 1:1 binding model and a binding constant (log $K_a$) of 6.5 ± 0.14, supporting our observation of a single quinol-binding site in ACIII.

ActC and ActF contain putative proton-conducting pathways (Fig. 6 and Supplementary Movie 2), which may allow proton translocation by ACIII to contribute to the electrochemical membrane potential. The channels are formed by residues with side chains that can establish hydrogen bonds, as those observed for other respiratory complexes such as complex I and HCOs[27–29]. This would allow proton transfer through a hydrogen bond network by successive breaking and concomitant establishment of hydrogen bonds, known as a Grotthuss mechanism[30,31]. In ActC, two half-channels formed by conserved residues are identified

(Fig. 6a and Supplementary Fig. 9), as in the case of PsrC. The cytoplasmic half-channel is formed by residues in the first four-helix bundle (TMHs 2–5) and includes Asp197$^C$, Arg196$^C$, Asp191$^C$, Arg119$^C$, Glu450$^C$, Arg114$^C$, Glu122$^C$, Ser103$^C$, Thr100$^C$, Thr125$^C$, Ser180$^C$, Tyr176$^C$, His97$^C$, and His247$^C$. In the second four-helix bundle (TMHs 6–9), we find the periplasmic half-channel, which comprises residues Arg395$^C$, Glu394$^C$, Tyr270$^C$, Tyr323$^C$, Glu326$^C$, and Glu338$^C$.

In ActF, we observe less strongly conserved residues defining similar cytoplasmic and periplasmic half-channels in the first and second four-helix bundles, respectively (Fig. 6b and Supplementary Fig. 11). Some of the residues in these two half-channels are Asp184$^F$, Arg100$^F$, Ser180$^F$, Glu103$^F$, His88$^F$, Ser80$^F$, Ser238$^F$, His355$^F$, Asp358$^F$, Tyr233$^F$, Tyr290$^F$, Arg308$^F$, and Glu301$^F$.

The observation of two likely proton pathways suggests the existence of two proton-pumping sites in ACIII. The presence of such pathways in homologous subunits is reminiscent of

respiratory complex I, where three proton-pumping sites have been suggested in the three homologous antiporter-like subunits[29].

ActD is the major contributor to the small cytoplasmic domain of ACIII, where both its N- and C-terminus are located. The subunit is composed of two TMHs that cross each other in a helix dimer, and a βαβ motif in both termini that together form an antiparallel four-stranded β-sheet (Supplementary Fig. 4b). ActD is located next to the first four-helix bundle of ActC, establishing contacts with TMHs 4 and 5 from this subunit.

The periplasmic loop connecting the two helices contains several conserved residues (Supplementary Fig. 13) and interacts with ActB. In addition, a highly conserved glutamate (Glu122[D]; 98% conservation) is present in the second TMH of ActD. Glu122[D] faces ActC and neighbors conserved residues from the other membrane subunits, including Asp169[C], Ser245[C], and Tyr284[F] (98–100% conservation; Fig. 7 and Supplementary Movie 2). Although the absence of electron density for aspartate and glutamate side chains in cryo-EM maps introduces a level of uncertainty in the modeling of Glu122[D] and Asp169[C], the latter, part of the quinol-binding site (Fig. 5), is found only ~8 Å away from Glu122[D]. The short distance between these two residues suggests that protonation or conformational changes of Asp169[C], linked to quinol oxidation, might be sensed by Glu122[D].

**ACIII–caa₃ supercomplex.** Extensive 3D classification revealed a small subset of particles with a density adjacent to the transmembrane region of ACIII (Supplementary Fig. 1c). The refinement of this subset yielded a 20 Å map with a shape suggestive of a supercomplex of ACIII and the caa₃ oxygen reductase, which had been predicted from BN-PAGE analysis[8]. Accordingly, we docked our ACIII model and the caa₃ oxygen reductase structure from *T. thermophilus* (PDB 2YEV)[32] into this map, which proved to accommodate the two complexes well (Fig. 2).

The fitting locates the large membrane subunit I/III from the caa₃ oxygen reductase next to ActC and below ActA, while the periplasmatic domain of subunit IIc of the terminal oxidase, which contains the primary entry point of electrons, is adjacent to the cytochrome subunits of ACIII (Fig. 2). Interestingly, it was previously shown that ActE is a direct electron donor to the caa₃ oxygen reductase[20]. Our low-resolution map of the ACIII–caa₃

assembly shows that the relative position of ActE and subunit IIc in the supercomplex is favorable for direct electron transfer between the two complexes, suggesting that their physical association serves a functional role.

## Discussion

ACIII is a quinol:cytochrome c/HiPIP oxidoreductase. All prosthetic groups in the complex are coordinated by the peripheral subunits ActA, ActB, and ActE, located in the periplasm. The prosthetic groups form two electron transfer wires that diverge at FeS1. The position of this iron–sulfur cluster, close to the quinol-binding site in ActC, implies FeS1 as the primary electron acceptor from the quinol. Reduction of the hemes from both ActA and ActE upon quinol oxidation has been previously reported[8], which supports the role of the heme wire in the electron transfer to the electron acceptor. The location of FeS4 near the protein surface raises the possibility that the [4Fe–4S] cluster wire might be also functional, receiving or donating electrons to an external electron carrier (Fig. 8). The amino acid residues composing the respective binding motifs are fully conserved. However, since the characterization of the [4Fe–4S] clusters has not been possible to date[3], the way in which these contribute to the electron flow in the complex remains elusive.

In several organisms, including *R. marinus*, a gene cluster coding for an oxygen reductase is found downstream of the cluster encoding ACIII[7,8,14]. The association of respiratory complexes into large macromolecular assemblies has been extensively studied in the last few years, but the physiological relevance of these so-called supercomplexes is still under debate[33]. Our data provides now structural evidence for the existence of an ACIII–caa₃ supercomplex in the membranes of *R. marinus*, previously biochemically and functionally characterized[8]. In this supercomplex, the last cofactor in the ACIII heme wire, present in ActE, is brought close to the cytochrome c domain of subunit IIc, the electron entering point of caa₃ oxygen reductase. Even though the accuracy of fit in a 20 Å map does not allow the determination of minimum distances between the two hemes, their orientation is compatible with the direct reduction of caa₃ oxygen reductase by ActE, as we previously proposed[20]. A recent study of the ACIII-aa₃ supercomplex from *Flavobacterium johnsoniae* also supports a direct electron transfer between the

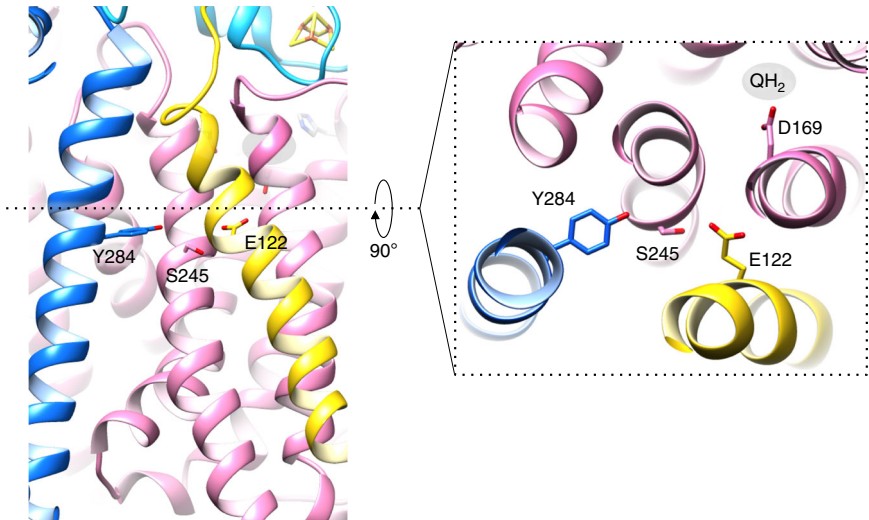

**Fig. 7** Inter-subunit cross-talk. Interface between ActC, ActF, and ActD in ACIII, as seen from the membrane (left) and from the periplasm (right). The highly conserved Glu122[D], Ser245[C], and Tyr284[F] are aligned in the central plane of the membrane, and are each 3–5 Å apart from their closest neighbor. Glu122[D] is ~8 Å from Asp169[C], which is part of the putative quinol-binding site. Subunits colored as in Fig. 1. Density for aspartate and glutamate residues is not visible and their side chains have been modeled

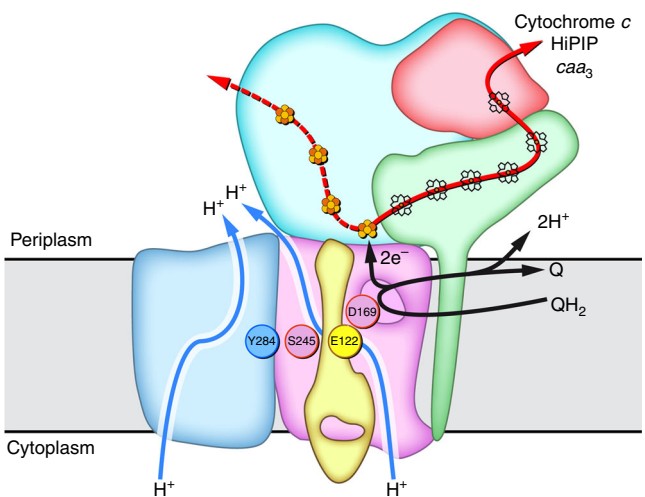

**Fig. 8** ACIII as a redox-driven proton pump. Proposed coupling between electron transfer and proton translocation. Upon quinol oxidation (black arrows), electrons are transferred to FeS1 and subsequently through the electron wires (red arrows); electrons conducted through the heme wire reduce a soluble electron carrier such as cytochrome $c$ or HiPIP, or the $caa_3$ oxygen reductase, as observed before[8, 10, 11]; clusters FeS2–4 may be involved in further electron transfer (red dashed arrow). Concurrently, a protonation or conformation change of Asp169[C] at the quinol-binding site may propagate through the conserved, neighboring residues Glu122[D], Ser245[C], and Tyr284[F] (colored circles). The conformational changes of these residues, located in one plane at the end of the putative proton half-channels in ActC and ActF, will allow further conformational changes at the proton-conducting channels for proton transport across the membrane (blue arrows). ActH was not included for clarity

complexes[58]. Interestingly, in this organism the architecture of the supercomplex is different from the one in *R. marinus* and ActA contains an extra cytochrome $c$ domain that appears to functionally replace the cytochrome $c$ domain from subunit IIc of the $caa_3$ oxygen reductase. The ACIII-oxygen reductase supercomplexes are, to this date, the only ones where the physical association of the complexes seems to have a clear functional role by bypassing the soluble electron carriers. The low resolution attained for the ACIII–$caa_3$ supercomplex is the consequence of both the reduced number of particles and, most likely, a weak and flexible binding that has been shown to be inherent to other respiratory supercomplexes[34,35]. Use of mild detergents for a stable isolation of the ACIII–$caa_3$ supercomplex may be useful in future structural studies.

Only one quinol-binding site was identified in ACIII (Fig. 5 and Supplementary Movie 2), contrary to the $bc_1$ complex, which has four quinone-binding sites in its functional dimeric form. This, together with the absence of any cofactors in the membrane, reinforces the notion that ACIII operates according to a mechanism totally distinct from the Q-cycle of the $bc_1$ complex and raises questions on how the complex contributes to the transmembrane difference of electrochemical potential. The existence of putative proton pathways in ActC and ActF (Fig. 6 and Supplementary Movie 2) suggests the operation of a redox-driven proton translocation mechanism in ACIII. These pathways are present in separated homologous subunits, mirroring the organization observed in complex I, where three putative proton pathways are present in three homologous subunits, the so-called antiporter-like subunits[36]. Indeed, this is not the only parallel that can be drawn between ACIII and complex I. Both enzymes are L-shaped complexes with a peripheral arm harboring an electron transfer chain and a membrane arm containing the proton-

pumping machinery. Although the mechanism is unknown, quinol reduction/oxidation would trigger conformational changes that initiate proton pumping in both complexes.

A fundamental question for these redox-driven pumps is: how is the redox reaction coupled to the translocation of protons across the membrane? In ACIII, the small membrane subunit ActD seems to be the key element for the articulation of the two components of the reaction. Located in the inner TMH of ActD, Glu122[D] is near Asp169[C] from the quinol pocket and, in this way, in an ideal position to sense protonation or conformational changes that might take place upon quinol oxidation. Interestingly, Glu122[D] is placed together with the conserved residues Ser245[C] and Tyr284[F] in a plane parallel to the membrane (Fig. 7 and Supplementary Movie 2) that coincides with the position at which the proton half-channels in ActC and ActF converge. We hypothesize that these residues are responsible for the coupling of quinol oxidation to proton translocation (Fig. 8). Such a mechanism will involve conformational changes triggered by redox activity and be propagated by the cross-talking residues Glu122[D], Ser245[C], and Tyr284[F] to the proton channels in ActC and ActF. Conformational changes at the channels will ultimately allow proton conduction, possibly by a Grotthuss-type mechanism[31,37].

Our study provides insights into the catalytic mechanism of a type of quinol:cytochrome $c$/HiPIP oxidoreductase. The structure corroborates the modular character of ACIII and its role as a redox-driven proton pump. According to thermodynamics, the oxidoreduction reaction involves the electron transfer of two electrons from menaquinol, with a reduction potential of −70 mV, to an electron acceptor with a reduction potential of +250 mV, which provides enough energy for pumping up to four protons[4,11,38]. The mechanism proposed here identifies the possible players responsible for quinol oxidation, proton translocation, and subunit crosstalk, and thus paves the way for testable hypotheses on the energy transduction mechanism of this complex.

## Methods

**Protein purification.** *R. marinus* strain PRQ-62B growth was performed as described before and ACIII was purified according to optimized procedures[3,13]. Briefly, *R. marinus* membranes were solubilized overnight at 4 °C in 20 mM Tris-HCl pH 8, 1 mM PMSF, 1 mM EDTA, and 2% DDM. Solubilized membranes were applied to a Q-Sepharose High Performance column. The sample was eluted applying a gradient of 0–500 mM NaCl in 20 mM Tris-HCl pH 8, 1 mM PMSF, 1 mM EDTA, and 0.05% DDM. The ACIII fraction, eluted with ~350 mM of NaCl, was then applied to a chelating Sepharose fast flow column saturated with Ni²⁺ and equilibrated with 20 mM Tris-HCl pH 8, 400 mM NaCl, and 0.05% DDM. The ACIII fraction was eluted in a linear gradient of 125 mM imidazole from 0 to 10% and then applied to a Q-Sepharose column. In this column, ACIII was eluted in 20 mM Tris-HCl pH 8 and 0.05% DDM with a linear gradient of 0–500 mM NaCl. The fraction containing ACIII was finally purified in a Superdex 200 column, eluted with 20 mM Tris-HCl pH 8, 150 mM NaCl, and 0.05% DDM. ACIII fractions were concentrated on VivaSpin 100 kDa concentrators and analyzed by SDS-PAGE and Blue Native-PAGE. ACIII aliquots were stored at −20 °C and thawed immediately before preparation of EM grids.

**Negative staining and initial model generation.** ACIII sample at 0.025 mg ml⁻¹ was negatively stained with 1% (w/v) uranyl acetate, pH ~4. Electron micrographs were recorded on a CCD camera (Gatan Ultrascan 4000) with a Tecnai Spirit at 120 kV under low-dose conditions, at a magnification of ×51,000 corresponding to a 2.34 Å pixel size at the specimen. Approximately 2000 particle images were picked manually in EMAN boxer[39] and used to generate templates. In total, 25,848 particle images were autopicked and used for two-dimensional reference-free classification in RELION 1.4[40]. The 2D class averages with recognizable features were selected and used to generate a low-resolution initial model with EMAN2[41].

**Single-particle cryo-EM data collection.** An aliquot of 3 μl of an ACIII sample at 1 mg ml⁻¹ was applied to freshly glow discharged C-Flat multihole holey carbon grids (Electron Microscopy Sciences). Grids were blotted for 9 s at 90% humidity and 10 °C in an FEI Vitrobot plunge freezer. Cryo-EM images were collected on a FEI Titan Krios operating at 300 kV aligned as described[42]. The microscope was

equipped with a Gatan K2 Summit electron detector and an energy filter. Images were recorded manually in counting mode at a nominal magnification of ×135,000, yielding a pixel size of 1.035 Å at the specimen. Movies were collected for 8 s with a total of 40 frames and a calibrated dose of about 1.8 $e^-$ $Å^{-2}$ per frame (total dose 72 $e^-$ $Å^{-2}$), at defocus values between −0.6 and −4.0 μm.

**Image processing and model building**. A set of 2479 movies was collected. Whole-image drift correction and dose weighting of each movie were performed using MotionCor2[43]. Particles were picked manually using EMAN Boxer[39] or automatically by template matching in Gautomatch (by Kai Zhang, MRC-LMB Cambridge, UK), and the micrograph-based CTF was determined using CTFFIND4 on drift-corrected, non-dose-weighted images[44]. Automatically picked particles were subjected to a first round of reference-free two-dimensional classification with ISAC within Sphire, to exclude false positives[45]. The initial clean data set contained 131,995 particle images (288 pixels × 288 pixels). Dose-weighted particles were subjected to 2D classification in RELION 1.4[40]. Visual selection of particle classes with interpretable features resulted in a data set of 103,756 particle images for 3D classification. The initial ACIII map generated with EMAN2 from negative-stained specimens was low-pass filtered to 60 Å and used as an initial reference for the 3D classification in RELION 1.4. The best 3D classes were selected for 3D refinement in RELION 2.0. Individual frames were B-factor weighted and movements of individual particles were reversed by movie frame correction in RELION 2.0[46]. The resulting data set of polished particles was used for a new 3D refinement, producing a final map at 3.87 Å resolution. Particles from 3D classes with a weak density at the membrane level were re-extracted with a larger box (300 pixels × 300 pixels) and extensively classified in 3D. The refinement of the final pool of particles with the extra density resulted in a 20 Å map. UCSF Chimera[47] was used for visualizing cryo-EM maps and docking of atomic models to the 20 Å map (ACIII from this work and *T. thermophilus caa₃* oxygen reductase—PDB 2YEV[32]). Model building was performed with Coot[48] and real space refinement was performed in Phenix[49] followed by manual rebuilding. MolProbity[50] and EMRinger[51] were used for model validation. Local resolution was estimated using ResMap[52]. Figures were drawn with UCSF Chimera.

**Peptide mass fingerprinting**. Identification of proteins present in the ACIII sample was performed using previously published protocols[53]. In brief, denatured, reduced, and alkylated proteins were digested using single and combinatorial digests of multiple proteases (Chymotrypsin, Lys-C and Trypsin) and desalted using $c_{18}$-ZipTips.

Proteolytic digests were loaded using a nano-HPLC (Bruker nanoElute) on reverse-phase columns (trapping cartridge: particle size 5 μm, C18, $L = 5$ mm (Thermo Fisher Scientific, Bremen); analytical column: particle size 1.9 μm, C18, $L = 40$ cm (Bruker Daltonik, Bremen)), and eluted in gradients of water (buffer A: water with 0.1% formic acid) and acetonitrile (buffer B: acetonitrile with 0.1% formic acid). All LC–MS-grade solvents were purchased from Fluka. Typically, gradients were ramped from 2 to 35% B in 180 min at flow rates of 300 nL $min^{-1}$. Peptides eluting from the column were ionized online in a Bruker CaptiveSpray ESI-source and analyzed in a Bruker Impact-II mass spectrometer. Mass spectra were acquired over the 150–2200 $m/z$ mass range. Sequence information was acquired by a computer-controlled, data-dependent, dynamic method with a fixed cycle time of 3 s and intensity-dependent acquisition speed for MS/MS-spectra between 8 and 20 Hz of the candidate ions. For sequential calibration 1 μl of calibrant (5 mM sodium formate in a 1:1 (v/v) mixture of $H_2O$ and acetonitrile) was injected for 2 min at the end of each chromatographic run. Each data file was recalibrated using the average spectrum from this time segment by Data Analysis (Bruker) and then exported in the mgf-format. The mgf files were processed using the Thermo Proteome Discoverer 2.2 software package. The derived mass lists were matched against the *R. marinus* Uniprot-database (downloaded from http://www.uniprot.org in 07/2017) and proteins were identified based on a 1% FDR using SEQUEST (Thermo Fisher Scientific, Bremen). The following search parameters were used: 10 ppm mass tolerance for precursor ions and 0.02 Da for fragment spectra; a total of two missed cleavages permitted for semi-tryptic peptides; oxidation of methionine and N-terminal acetylation as variable modifications and fixed carbamidomethylation of cysteine. PSM validation was performed by Percolator. Identified proteins were validated using the Mascot and MaxQuant software packages. All proteomic data associated with this study have been deposited to the ProteomeXchange Consortium via the PRIDE partner and is available using the accession number PXD008247.

**Isothermal calorimetry**. All ITC experiments were performed on a Micro-200 ITC (MicroCal, Malvern) at 65 °C, the optimal growth temperature of *R. marinus*. ACIII was buffer exchanged into ITC buffer 20 mM KPi, 10% glycerol, and 0.05% DDM with a PD-10 desalting column (GE Healthcare). ACIII was diluted to 19 μM, filtered, degassed, and introduced into the sample cell. The injection syringe was loaded with 190 μM DMN. ITC experiments were initiated by a 0.5 μL, followed by 3 μL injections every 600 s, with a stirring speed of 400 rpm. Titrations with DMN in the absence of ACIII were performed as controls. The protein activity was checked after ITC experiments as an additional control and shown to be

unchanged. The integration of thermograms was carried out with default NITPIC parameters.

**Bioinformatics**. Gene cluster coding for ACIII (*actABCDEF* or *actAB1B2CDEF*) was searched with the protein BLAST (BLASTp) analysis tool running at KEGG (Kyoto Encyclopedia of Genes and Genomes) database platform[54–56], as described[1,12]. The sequences from *R. marinus* were used as templates. Multiple sequence alignments were performed in CLUSTALW. The aligned sequences were used to generate a sequence logo using WebLogo 2.8.2[57].

**Data availability**. Data supporting the findings of this manuscript are available from the corresponding authors upon reasonable request. The cryo-EM map of ACIII was deposited in the Electron Microscopy Data Bank with accession code EMD-4165 and the structure coordinates were deposited in the Protein Data Bank with accession number 6F0K. All proteomics data associated with this manuscript have been deposited with the PRIDE online repository under the accession code PXD008247.

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

## Acknowledgements

We thank Margarida Bastos for discussions on ITC experiments and Imke Wüllenweber and Fiona Rupprecht for excellent technical assistance. F.C. is recipient of a fellowship by Fundação para a Ciência e a Tecnologia (PD/BD/104481/2014). The work was funded by Fundação para a Ciência e a Tecnologia (IF/01507/2015 to M.M.P.) and the Max Planck Society. The project was supported by LISBOA-01-0145-FEDER-007660 co-funded by FEDER through COMPETE2020-POCI and by Fundação para a Ciência e a Tecnologia. Support by FCT—Fundação para a Ciência e a Tecnologia and by UID/MULTI/04046/2013 center grant from FCT, Portugal (to BioISI) and the COST Action CM1306—Understanding Movement and Mechanism in Molecular Machines is also acknowledged.

## Author contributions

J.S.S. performed cryo-EM work and imaging processing. F.C. performed protein purification and ITC experiments. J.D.L. performed and analyzed mass spectrometry experiments. D.J.M. optimized electron microscope alignment and assisted cryo-EM work. J.S.S. and J.V. determined the structure. F.C. and P.N.R. performed sequence alignments. J.S.S., F.C., J.V. and M.M.P. interpreted the data and wrote the manuscript. M.T., W.K., J.V. and M.M.P. directed the project. All authors discussed the results and commented on the manuscript.

## Additional information

**Competing interests:** The authors declare no competing interests.

