## [Peer Review File · Nature Communications]

Reviewers' comments:

Reviewer #1 (Remarks to the Author):

The manuscript describes the first structure of respiratory alternative complex III, determined by cryo-EM. Its architecture is completely different from other known respiratory proton pumps, so the manuscript will be of high interest to a wide audience. The results are novel and the data is technically sound. The manuscript is well written and conclusions are supported by data and illustrations in most cases. Some attention is needed to points listed below.

Comments:

Very often authors use words "new protein", "new subunit", "new chimera" etc. In many cases this looks a bit strange and the use of "new" should be pared down. Or at least changed to "new type of protein, etc" if left in.

The resolution for ACIII.caa3 supercomplex (~20 Å) seems a bit low for ~10K particles at these imaging conditions. One could expect perhaps about 6 Å – did authors try to investigate reasons for that? Can this class be classified further into more homogeneous population? Was focused refinement of just caa3 attempted?

Figure 4 – Subunits should be indicated.

Figure 5 – quinone cavity should be shown in surface representation.

Line 135 – symmetry-related domains should be illustrated in the Supplement.

Line 185 – identification of ActH is not quite convincing as described. Why authors had > 150 candidate proteins? Presumably in their purified sample all the bands on SDS-PAGE, apart from one, should have been assigned before? The MS of unidentified band then should give just one-two candidates with high abundance and corresponding MW. Was such an experiment performed? If not, it should be. The details should be illustrated with SDS-PAGE of the purified sample, protein bands identified and a MS table for identification. Lacking such characterization authors rely on cryo-EM density to confirm their identification but Fig. 3E is not quite convincing for that. Perhaps longer stretches of density with clear bulky residues should be shown.

Suppl. Fig. 5C is too small to read – should be a separate figure.

Figure 6 – from this depiction it is not clear how close to protein surface the discussed residues are and whether they are exposed within internal cavities. Surface representation, including internal cavities, will help. This is especially important because authors propose several Arg residues in channels – they normally cannot be de-protonated but do carry a charge, so are not good candidates for the deep intra membrane part of the proton "wire".

Line 320 – are residues coordinating FeS clusters fully conserved in ACIII? If some are not, this may suggest that clusters are just structural, remaining from an ancestor where they may have been functional?

Line 371 – are Glu122D, Ser245C and Tyr284F fully conserved? How exposed (or not) is Glu122D – seems to be in a too peripheral position for the key role?

Reviewer #2 (Remarks to the Author):

In this manuscript the authors claim to have generated a structure of alternative complex III (ACIII) from *Rhodothermus marinus* to a resolution of 3.9 Å using cryo-EM. The complex has 7 subunits and 6 of these are identified by and fitted into electron density. Subunit ActG representing 130 residues is missing and an additional subunit 'ActH' has been identified as part of the complex. Unexplained density is also seen as part of the complex. The overall structure of the assembly is defined and in so doing residues responsible for proton conduction and FeS centers responsible for electron transfer have been tentatively identified. The putative substrate quinol binding pocket has been located and a means for coupling between electron transfer and proton pumping has been proposed. Within the population of particles examined a fraction has been described as consisting of a supercomplex between ACIII and a terminal oxidase in the respiratory chain.

Whilst ACIII was discovered almost two decades ago and is not new, the structure itself is. As a novel structure, it will be of interest to others in the community and in light of it representing a 'minimal' complex III it will be of interest to the wider field. Indeed, the structure suggests many testable hypothesis that should lead to additional research effort in the area. The supercomplex is particularly intriguing and, if confirmed will attract considerable attention and will influence thinking in the field.

Is the work convincing? In its current form it is not. The materials and methods section is woefully incomplete. No information is provided about the protein sample used for cryo-EM. The procedures implemented for sample preparation up to and including cryo-EM are not described. Reference is made to the fact that particles were randomly oriented. Data in support of this statement should be included especially given the L-shape of ACIII. There are other major and minor issues itemised in the list that follows that must be responded to.

26. 34, and elsewhere. ACIII was already described in 1999. It is not new.

28. State that the resolution is 3.9 Å. The reader can decide as to whether or not this is high-resolution.

31. The quinol site location was not established.

45. Not only prokaryotes. Eukaryotes too have alternative complexes. NdiI in yeast is one such example.

48. Statement makes no sense.

52. 1999 is not recent.

58. Some attempt at rationalizing the presence of both ACIII and complex III should be provided.

64. Evidence for supercomplex formation is required.

79. ACIII is not a chimera.

84. The % residues in useful density should be included here and in the abstract.

91. The reader will expect to see reference to the DDM micelle in the final structure. This should be addressed.

93. The randomness of orientation can be quantified and should be reported. It is hard to imagine that an L-shaped particle does not preferentially orientate.

99. The composition of the sample used for cryo-EM must be described in detail. The assorted components, including *caa3*, must then be evaluated as possible components of the proposed supercomplex.

101. Since EM alone does not inform orientation in the membrane, independent support for the particular orientation should be referred to.

111. Indicate what exactly was subjected to fingerprinting and how it was isolated.

121. State the number of crossings for ActG. Indicate if ActG copurifies as part of ACIII.

132. Show figure to illustrate how weak the density actually is.

133. ACIII is an iron rich protein. Describe if/how radiation damage was quantified as part of the cryo-EM data collection and structure determination process.

170. The authors should speculate on the role of the FeS2 – FeS4 branch in ActB.

178. Rewrite this sentence for clarity.

182. Presumably the lipidation at the N-terminus of ActE is missing in the electron density. This should be referred to and accounted for in the electron density tally.

189. This raises questions regarding sample purity.

201. Precedence for the stability statement should be included.

220. Show elongated densities and where they are located in the complex. Is the DDM micelle seen?

228. A structure with quinol or an analog bound should be included as part of the study. It is an obvious experiment to do and would dispense with much of the speculation in the manuscript. A binding constant should be included too.

256. The criteria used for identifying the proton conduction pathway should be included. The proposed pathways should be shown in a separate figure with distances and chemistries clearly defined. Explain how the Grotthus mechanism would apply. Molecular dynamics simulations should be run to evaluate the feasibility of the proposed pathways. Indicate how the proposed pathways compare with other well-established pathways.

291. Include evidence in support of the ACIII-*caa3* supercomplex. Cross-linking, pull-down, BN gel, gel chromatography, stoichiometry, etc.

305. Show figure with distances and orientation of electron donors and acceptors in the supercomplex. Describe the shape, charge complementarity between interacting surfaces.

328. At 20 Å resolution, the evidence for the existence of this particular supercomplex is not at all convincing.

337. Explain why bypassing soluble carriers is interesting, relevant.

354. Are ACIII and complex I the only L-shaped complexes?

501. Many of the figures are hard to read and should be properly and clearly labelled.

509. Include a resolution color-coded map of ACIII.

Fig. 2c. Point out the line between donor and acceptor between ACIII and *caa3* in the supercomplex

Hard to see what surfaces are in contact in the supercomplex.

Explain how the cleft in the putative membrane between ACIII and *caa3* is stabilized.

Explain what, other than *caa3*, in the sample used for cryo-EM could account for the extra density and the quality of that fit.

Fig. 3B is confusing.

528. Label residues.

534. Label subunits for clarity.

State error estimates on distances at 3.9 Å resolution. Is this uniform across all electron and proton conducting partners?

Comment on the state of oxidation/reduction of the irons in FeS and hemes under data collection conditions during cryo-EM. Does this impact on the observed/reported structure? This ties in with the issue of radiation damage.

536. In the absence of a complex with bound quinol or an analog, quinol should be docked into the proposed binding pocket and the result discussed.

Fig. 6. Label residues and helices. See comment for line 305. Add membrane boundaries.

Fig. 7. Show where these residues are in Fig. 6.

561. How reliable is this 8 Å value? It is a very long distance. Comment on the likelihood of communication over such a large distance.

Electron density should be added for all residues where distances are referred to and considered important in ACIII function.

Fig 8. Explain why electrons should exit through ActE and not via the other FeS centers in ActE. Show Asp169 in ActC in the figure.

Refer to ActG and ActH.

579-602. Full details needed for the reader to properly evaluate the work.

632. Details required.

666. 65 deg C is a very high temperature at which to study a protein. Indicate the functional stability of the protein under these conditions. How long is the protein held at this high temperature?

Fig. s2. Label subunits.

Show density around hemes, FeS centers and coordinating residues.

Fig. s3B. Unclear. Indicate that this is missing from the electron density and the complex structure

reported.

765. Explain how E122 is stabilized in the membrane interior.

Fig. s5C, s6, s7. Illegible.

Fig. s8. Report on the functional state of enzyme at the end of an ITC measurement at 65 C.

Table s1. List the 10 ligands.

Reviewer #3 (Remarks to the Author):

This manuscript presents a cryo-EM structure of Alternative Complex III, which is a relatively recently discovered electron transport complex found in many bacteria. It appears to be a functional replacement for the cyt bc1 complex (complex III), but structurally and evolutionarily unrelated.

The determination of a structure for this complex is exciting and I support publication of this landmark achievement. However, I do have some questions and suggestions that I think will improve the manuscript. Most importantly is the question of whether or not ACIII is really a proton-pumping complex. I am not aware of any experiments that show this. The manuscript seems to assume that this is true, but the structure doesn't really give much support to that idea. The putative proton transfer channels are far removed from the electron transfer chains, and while that can certainly be the case, as in Complex I, evidence is needed that such proton transfer chains exist. As far as I can tell, the structure is entirely compatible with a mechanism in which the quinone is oxidized and the two protons ejected to the periplasm, but with no other protons pumped. This might be a more primitive type of complex and the bc1 complex with the Q cycle that pumps protons could be an improved version.

Some discussion of the thermodynamics of the electron flow in the complex seems to be appropriate. The potential drop from the quinone to the terminal cytochrome acceptor seems to be relatively shallow, compared to either the bc1 complex or either Complex I or Complex IV. If proton pumping is going to take place, the energy to drive it ultimately has to come out of the redox gradient. Is there enough energy here to drive proton pumping?

We thank the reviewers for their detailed and, on the whole, positive comments that have helped us to improve the manuscript. Here is our detailed, point-by-point response.

Reviewer #1

The manuscript describes the first structure of respiratory alternative complex III, determined by cryo-EM. Its architecture is completely different from other known respiratory proton pumps, so the manuscript will be of high interest to a wide audience. The results are novel and the data is technically sound. The manuscript is well written and conclusions are supported by data and illustrations in most cases. Some attention is needed to points listed below.

Response

We thank Reviewer #1 for his/her critical review of the manuscript. We are happy the Reviewer finds “the manuscript will be of high interest to a wide audience” and that the “results are novel and the data is technically sound”.

Comment 1

Very often authors use words “new protein”, “new subunit”, “new chimera” etc. In many cases this looks a bit strange and the use of “new” should be pared down. Or at least changed to “new type of protein, etc” if left in.

Response

The manuscript was changed according to the Reviewer’s suggestion.

Comment 2

The resolution for ACIII.caa3 supercomplex (~20 Å) seems a bit low for ~10K particles at these imaging conditions. One could expect perhaps about 6 Å – did authors try to investigate reasons for that? Can this class be classified further into more homogeneous population? Was focused refinement of just caa3 attempted?

Response

The reason for the low resolution is the weak, flexible interaction of the two complexes in the supercomplex. The small dataset we used was the end result of an extensive classification procedure. Focused refinement was not performed, because the crystal structure of caa₃ is known and we were able to show its orientation in the map. The important information we are

missing is the detailed interaction between the two complexes; we plan a purification of the supercomplex to investigate this.

Comment 3

Figure 4 – Subunits should be indicated.

Response

Figure 4 was changed according to the Reviewer's suggestion.

Comment 4

Figure 5 – quinone cavity should be shown in surface representation.

Response

Movie 2 was added in the revised manuscript, where the quinol cavity is shown in surface representation as suggested by the Reviewer.

Comment 5

Line 135 – symmetry-related domains should be illustrated in the Supplement.

Response

Supplementary Figure 4a includes now the amino acid sequence of ActA with heme motifs colour-coded as in the structure, from which the pseudo-symmetric domains should be recognizable.

Comment 6

Line 185 – identification of ActH is not quite convincing as described. Why authors had > 150 candidate proteins? Presumably in their purified sample all the bands on SDS-PAGE, apart from one, should have been assigned before? The MS of unidentified band then should give just one-two candidates with high abundance and corresponding MW. Was such an experiment performed? If not, it should be. The details should be illustrated with SDS-PAGE of the purified sample, protein bands identified and a MS table for identification. Lacking such characterization authors rely on cryo-EM density to confirm their identification but Fig. 3E is not quite convincing for that. Perhaps longer stretches of density with clear bulky residues should be shown.

Response

The detection limit of an optimized bottom-up LC-MS/MS experiment is today in the range of attomoles of a protein. This exceeds visualization on gels or blots, and will also lead to the identification of trace amounts of contaminating proteins. Also, there is only limited correlation of protein abundance in a sample and measured peptide intensity in the mass spec. This is caused by presence (or absence) of suitable proteolytic cleavage sites to generate peptides in the first place and also by the ionization efficiencies of the respective peptides. Thus, the intensities observed in a mass spec cannot yield absolute, quantitative information about a protein's abundance. Thus, the intensities observed in a mass spec cannot yield absolute, quantitative information about a protein's abundance. Statistically, though, counting peptides and the respective intensities does allow estimations of protein abundance for large, soluble proteins. In the digests we performed, we used relatively large amounts of protein and analyzed the samples on high-end LC-MS/MS setups. We wanted to obtain as comprehensive an overview of the sample as possible, acknowledging the fact that this approach might lead to an "over-appearance" of small amounts of contaminating proteins. We then made use of a combination of cryo-EM data-derived size constraints and MS data-derived spectral counting and intensity estimations to pre-select a sub-set of proteins identified in the sample, and then identify the unknown subunit by matching secondary structure elements. We found that the protein hit Rmar_1979 (ActH) was in the mass range predicted from the cryo-EM data and had the highest number of unique peptides. It also displayed an excellent match in secondary structure elements and we were able to overlap bulky side chains with electron densities. No other proteins identified in the sample matched these criteria. We are thus confident that we have correctly identified the previously unknown subunit.

To better visualize the excellent agreement of the protein sequence with the cryo-EM data, we have added a supplemental movie showing the overlap between large side chains in the primary sequence and electron densities in the cryo-EM data (Movie 1).

Regarding the EM density, we have modified the text, line 238-240: " The good side chain densities in this region enabled us to trace Rmar_1979 in the map (Fig. 3d,e and Movie 1), and we thus assigned it as "ActH"."

Comment 7

Suppl. Fig. 5C is too small to read – should be a separate figure.

Response

Supplementary Figure 5C is now a separate figure (Supplementary Figure 7).

Comment 8

Figure 6 – from this depiction it is not clear how close to protein surface the discussed residues are and whether they are exposed within internal cavities. Surface representation, including internal cavities, will help. This is especially important because authors propose several Arg residues in channels – they normally cannot be de-protonated but do carry a charge, so are not good candidates for the deep intra membrane part of the proton “wire”.

Response

In order to address these points we added a new movie (Movie 2) to the revised manuscript.

Comment 9

Line 320 – are residues coordinating FeS clusters fully conserved in ACIII? If some are not, this may suggest that clusters are just structural, remaining from an ancestor where they may have been functional?

Response

Yes, the residues are fully conserved, as is now mentioned on line 397.

Comment 10

Line 371 – are Glu122D, Ser245C and Tyr284F fully conserved? How exposed (or not) is Glu122D – seems to be in a too peripheral position for the key role?

Response

Glu122D and Ser245C are conserved in 98 % of the analyzed sequences, while Tyr284F is 100 % conserved. In the two sequences in which Ser245C is not present there is another Ser three residues forward, i.e. one turn up in the same helix. Glu122D is not exposed as it can be observed for example in Figure 7 and Movie 2. Please notice that, in the figure, the observed helix from ActD has the second helix from this subunit in front of it from the observation point.

Reviewer #2

*In this manuscript the authors claim to have generated a structure of alternative complex III (ACIII) from *Rhodothermus marinus* to a resolution of 3.9 Å using cryo-EM. The complex has 7 subunits and 6 of these are identified by and fitted into electron density. Subunit ActG representing 130 residues is missing and an additional subunit ‘ActH’ has been identified as*

part of the complex. Unexplained density is also seen as part of the complex. The overall structure of the assembly is defined and in so doing residues responsible for proton conduction and FeS centers responsible for electron transfer have been tentatively identified. The putative substrate quinol binding pocket has been located and a means for coupling between electron transfer and proton pumping has been proposed. Within the population of particles examined a fraction has been described as consisting of a supercomplex between ACIII and a terminal oxidase in the respiratory chain.

Whilst ACIII was discovered almost two decades ago and is not new, the structure itself is. As a novel structure, it will be of interest to others in the community and in light of it representing a 'minimal' complex III it will be of interest to the wider field. Indeed, the structure suggests many testable hypothesis that should lead to addition research effort in the area. The supercomplex is particularly intriguing and, if confirmed will attract considerable attention and will influence thinking in the field.

Response

We thank Reviewer #2 for his/her critical review of the manuscript. We are pleased the Reviewer finds our data "will be of interest to others in the community" as to "the wider field" and importantly "the structure suggests many testable hypothesis that should lead to addition research effort in the area".

Comment 1

Is the work convincing? In its current form it is not. The materials and methods section is woefully incomplete. No information is provided about the protein sample used for cryo-EM. The procedures implemented for sample preparation up to and including cryo-EM are not described. Reference is made to the fact that particles were randomly oriented. Data in support of this statement should be included especially given the L-shape of ACIII. There are other major and minor issues itemised in the list that follows that must be responded to.

Response

We have modified the manuscript in order to include the information mentioned by the Reviewer. Information about the protein sample used for cryo-EM is now included in lines 470-491. A 3D plot of the angular distribution for the particles included in the final reconstruction is now shown in Supplementary Figure 1d. For the other issues itemised please refer to the list below.

Comment 2

26. 34, and elsewhere. ACIII was already described in 1999. It is not new.

Response

The term “new” was removed.

Comment 3

28. State that the resolution is 3.9 Å. The reader can decide as to whether or not this is high-resolution.

Response

The manuscript was changed according to the Reviewer’s suggestion.

Comment 4

31. The quinol site location was not established.

Response

The sentence has been changed (line 29):

“ACIII presents a so-far unique structure, for which we established the arrangement of the cofactors (four iron-sulfur clusters and six C-type hemes) and propose the location of the quinol-binding site and the presence of two putative proton pathways in the membrane.”

Comment 5

45. Not only prokaryotes. Eukaryotes too have alternative complexes. Ndil in yeast is one such example.

Response

The sentence was modified accordingly (line 50).

Comment 6

48. Statement makes no sense.

Response

The statement was removed.

Comment 7

52. 1999 is not recent.

Response

The manuscript was changed accordingly.

Comment 8

58. Some attempt at rationalizing the presence of both ACIII and complex III should be provided.

Response

A sentence addressing this issue was included in the manuscript, as suggested (line 64).

Comment 9

64. Evidence for supercomplex formation is required.

Response

We have previously characterized the association of ACIII and *caa*₃ oxygen reductase functionally and biochemically (Refojo *et al*, 2010 *Biochim Biophys. Acta*, 1797, 1477). Reference to this paper can be found in lines 69-72.

Comment 10

79. ACIII is not a chimera.

Response

The statement was removed.

Comment 11

84. The % residues in useful density should be included here and in the abstract.

Response

Counting the residue densities would not be objective, and there are no known metrics to judge what percentage would be acceptable in a 3.9 Å structure. We have used validation

methods that are standard practice in the cryo-EM community, as reported in Supplementary Table 1.

Comment 12

91. The reader will expect to see reference to the DDM micelle in the final structure. This should be addressed.

Response

Supplementary Figure 2a includes now an ACIII map at a low density value, where the detergent micelle is visible.

Comment 13

93. The randomness of orientation can be quantified and should be reported. It is hard to imagine that an L-shaped particle does not preferentially orientate.

Response

A 3D plot of the angular distribution for the particles used in the final reconstruction is now included in Supplementary Figure 1d. This plot shows that the particles in the final reconstruction are randomly oriented and Fourier space is sampled evenly.

Comment 14

99. The composition of the sample used for cryo-EM must be described in detail. The assorted components, including caa3, must then be evaluated as possible components of the proposed supercomplex.

Response

A detailed description of the sample used for cryo-EM is now included (lines 470-491). The biochemical and functional characterization of the supercomplex is described in Refojo *et al*, 2010 *Biochim Biophys. Acta*, 1797, 1477.

Comment 15

101. Since EM alone does not inform orientation in the membrane, independent support for the particular orientation should be referred to.

Response

Support for the particular orientation is now referred to in the manuscript, as suggested by the Reviewer (lines 124-126).

Comment 16

111. Indicate what exactly was subjected to fingerprinting and how it was isolated.

Response

The same sample used for cryo-EM was used in the mass spectrometry experiments (line 540).

Comment 17

121. State the number of crossings for ActG. Indicate if ActG copurifies as part of ACIII.

Response

ActG has one predicted TMH. This is now indicated in line 141. ActG co-purification was observed as it is illustrated by the SDS-PAGE gel included in Supplementary Figure 1a.

Comment 18

132. Show figure to illustrate how weak the density actually is.

Response

A figure to illustrate how weak the density is now included as Supplementary Figure 5.

Comment 19

133. ACIII is an iron rich protein. Describe if/how radiation damage was quantified as part of the cryo-EM data collection and structure determination process.

Response

Radiation damage was taken into account in the image processing procedure by dose-weighting the images using the particle polishing procedure in RELION, as described in the Methods section. This downweights the high-resolution information of the later, more radiation-damaged frames. There is no reason to think that the iron is especially sensitive to radiation damage by electrons. The FeS clusters and their cysteine ligands are well-defined in the map, as are the hemes. We show densities in new Supplementary Figures 2d and e. There

are high-resolution structures published of other iron-rich proteins, including respiratory complex I and the iron-nickel hydrogenase Frh. Most relevantly, a cryo-EM map of Frh at better than 3 Å resolution (EMD-3518 in the Electron Microscopy Data Bank (http://emsearch.rutgers.edu/atlas/3518_summary.html)) shows high-resolution detail of four different FeS clusters and a NiFe center.

Comment 20

170. The authors should speculate on the role of the FeS2 – FeS4 branch in ActB.

Response

The possible role of FeS2-FeS4 was considered in the Discussion section, lines 390-396.

Comment 21

178. Rewrite this sentence for clarity.

Response

The sentence was rewritten as "Electron density for ActE in the membrane is not visible, preventing further modeling." (lines 213-214).

Comment 22

182. Presumably the lipidation at the N-terminus of ActE is missing in the electron density. This should be referred to and accounted for in the electron density tally.

Response

Indeed we see no density for the lipidation at N-terminus of ActE. The lipid would only be visible in the map if it was well-ordered in the membrane environment, which is not expected. We refer to this in line 213-214.

Comment 23

189. This raises questions regarding sample purity.

Response

Please refer to our response to Reviewer #1 **comment 6**. We stress again that the LC-MS/MS approach employed here has an extremely low detection threshold. In total, 118 of the 151

proteins detected in the preparation had < 5 unique peptides and comprised only a fraction of the proteins present in solution.

Comment 24

201. *Precedence for the stability statement should be included.*

Response

We do not know of a specific precedent for this, and it would be very difficult to establish such a role definitely. We are merely speculating on the role of ActH, which has no cofactors and is not close to the active sites of the complex, but is tightly interacting with three neighbouring subunits. A role in stability appears plausible.

Comment 25

220. *Show elongated densities and where they are located in the complex. Is the DDM micelle seen?*

Response

The figure below illustrates one of the elongated densities observed in the map. Inspection of the deposited EM map might provide a better visualization of these densities.

The DDM micelle is observed. Please refer to our reply to **comment 12** above.

a, slice of ACIII density map, as seen from ActF (blue – ActF; pink – ActC; red – elongated density). **b**, segmented density consistent with lipid or detergent molecule. **c**, fit of an illustrative lipid (C₄₀H₇₉O₁₀P) into segmented density.

Comment 26

228. A structure with quinol or an analog bound should be included as part of the study. It is an obvious experiment to do and would dispense with much of the speculation in the manuscript. A binding constant should be included too.

Response

A structure with a bound quinol or analog would be interesting and we are seeking to obtain one. This is likely to take years and will be a separate study that goes well beyond the scope of our present manuscript. Nonetheless, our data and the fact that the proposed quinol pocket also agrees with the quinol-binding site of PsrC, identified by co-crystallization experiments, strongly supports the proposed location of the quinol pocket in ACIII. A binding constant is now included as suggested (line 315 and legend of Supplementary Figure 10).

Comment 27

256. The criteria used for identifying the proton conduction pathway should be included. The proposed pathways should be shown in a separate figure with distances and chemistries clearly defined. Explain how the Grotthuss mechanism would apply. Molecular dynamics simulations should be run to evaluate the feasibility of the proposed pathways. Indicate how the proposed pathways compare with other well-established pathways.

Response

The criteria used for identifying the proton conduction pathways are now included in the manuscript as suggested by the Reviewer (lines 320-322). We added another movie (Movie 2) to the revised manuscript, which illustrates the proposed pathways. A brief explanation of the Grotthuss mechanism is now given (lines 322-325), as well as, how the proposed pathways compare with other well-established pathways (lines 320-322). At the obtained resolution molecular dynamics simulations would be speculative and would not add much to our conclusions.

Comment 28

*291. Include evidence in support of the ACIII-*caa3* supercomplex. Cross-linking, pull-down, BN gel, gel chromatography, stoichiometry, etc.*

Response

Please refer to our previous publication, Refojo *et al*, 2010 *Biochim Biophys. Acta*, 1797, 1477 where BN gels and functional data are reported.

Comment 29

305. Show figure with distances and orientation of electron donors and acceptors in the supercomplex. Describe the shape, charge complementarity between interacting surfaces.

Response

An arrow indicating the electron transfer between ACIII and *caa*₃ was added to the figure indicating the relative orientation of electron donors and acceptors in the supercomplex. At the resolution obtained it would be speculative to quantify such distances or discuss the charge complementarity as requested.

Comment 30

328. At 20 Å resolution, the evidence for the existence of this particular supercomplex is not at all convincing.

Response

Please refer to our previous work, Refojo *et al*, 2010 *Biochim. Biophys. Acta*, 1797, 1477, where biochemical and functional data are reported.

Comment 31

337. Explain why bypassing soluble carriers is interesting, relevant.

Response

We have previously observed that the monoheme cytochrome *c* subunit (ActE) of ACIII is a direct electron donor to *caa*₃ oxygen reductase in *R. marinus* (Refojo *et al* 2017, *Biol. Chem.* 398, 1037). In the referred sentence we just mentioned that the structure here obtained is compatible with that observation.

Comment 32

354. Are ACIII and complex I the only L-shaped complexes?

Response

Indeed complex I (and related complexes) and ACIII are the only L-shaped respiratory complexes we know of.

Comment 33

501. *Many of the figures are hard to read and should be properly and clearly labelled.*

Response

Figures were redrawn as suggested.

Comment 34

509. *Include a resolution color-coded map of ACIII.*

Response

This is now included as Supplementary Figure 1f.

Comment 35

Fig. 2c. Point out the line between donor and acceptor between ACIII and caa_3 in the supercomplex. Hard to see what surfaces are in contact in the supercomplex. Explain how the cleft in the putative membrane between ACIII and caa_3 is stabilized. Explain what, other than caa_3 , in the sample used for cryo-EM could account for the extra density and the quality of that fit.

Response

As mentioned above, an arrow was included in the figure indicating the relative orientation of electron donors and acceptors in the supercomplex. The interaction between ACIII and caa_3 might be stabilized by other interaction partners not included in the fit, as for example the ActG subunit whose location in the complex is still unknown. Lipid-protein interactions can also play a role in the formation of the supercomplex.

We cannot hypothesize what, other than caa_3 , could account for the extra density. We proposed the extra density to be the caa_3 based on our previous data (Refojo *et al*, 2010 *Biochim Biophys. Acta*, 1797, 1477) and on gene clustering information since in *R. marinus* the gene cluster coding for the caa_3 is preceded by that encoding ACIII.

Comment 36

Fig. 3B is confusing.

Response

The figure has been redrawn to make it clearer. We ask for this figure to be a two-column figure for a clear visualization of the labels in the MS/MS spectrum.

Comment 37

528. Label residues.

Response

Residues were labeled as suggested.

Comment 38

534. Label subunits for clarity.

Response

Subunits were labeled as suggested.

Comment 39

State error estimates on distances at 3.9 Å resolution. Is this uniform across all electron and proton conducting partners?

Response

Distances between centers of density, as FeS clusters or heme irons, are very accurate and not much dependent on resolution. Because we fit atomic models, the edge-to-edge distances between hemes can also be measured with confidence. Problematic are glutamate and aspartate side chains, which are very radiation-sensitive and usually not seen in cryo-EM maps, whereas other side chains in the protein interior are mostly well-defined. Throughout the text we have opted for terms like "less than 6 Å" or "~4 Å" where appropriate, in order not to suggest an unrealistic accuracy.

Comment 40

Comment on the state of oxidation/reduction of the irons in FeS and hemes under data collection conditions during cryo-EM. Does this impact on the observed/reported structure? This ties in with the issue of radiation damage.

Response

We cannot comment on the oxidation state of the irons, but we have no reason to believe that this impacts the observed structure. The environment of the hemes and FeS clusters are among the best-resolved regions of the map. Please refer to our response to **comment 19**.

Comment 41

536. In the absence of a complex with bound quinol or an analog, quinol should be docked into the proposed binding pocket and the result discussed.

Response

As mentioned above, we are seeking to obtain the structure of the complex with bound quinol or an analog (please refer to our response to **comment 26**). Also, as mentioned before performing docking experiments at this resolution would be speculative.

Comment 42

Fig. 6. Label residues and helices. See comment for line 305. Add membrane boundaries.

Response

Residues were labeled and membrane boundaries added. Since the two four-helix bundles are identified as the structural units of each half-channel, the TMHs were not individually labeled for reasons of clarity.

Comment 43

Fig. 7. Show where these residues are in Fig. 6.

Response

For reasons of clarity, the information in Figures 6 and 7 should not overlap. Instead, we added a new movie (Movie 2) that shows the positions of the residues discussed in the text.

Comment 44

561. How reliable is this 8 Å value? It is a very long distance. Comment on the likelihood of communication over such a large distance.

Electron density should be added for all residues where distances are referred to and considered important in ACIII function.

Response

In cryo-EM maps no density is typically observed for the side chains of aspartates and glutamates due to radiation damage that induces decarboxylation of these residues very early in the imaging process. For this reason, the location of the side chains of Asp169C and E122D cannot be accurately determined; the 8 Å distance is approximate, as indicated in line 361. Water molecules might mediate the communication over this distance, and also conformational changes might occur upon quinol binding/oxidation.

As mentioned in our response to **comment 11**, we have used well-established validation methods and since density cannot be shown for several of the residues implicated in ACIII function as explained above we think inspection of the map and model deposited in the PDB data bank should address better the second point raised by the reviewer.

Comment 45

Fig 8. Explain why electrons should exit through ActE and not via the other FeS centers in ActE. Show Asp169 in ActC in the figure.

Refer to ActG and ActH.

Response

The flow of the electrons is discussed in lines 392-400. Previous work supports the role of the hemes in the electron transfer (Refojo *et al*, 2010 *Biochim. Biophys. Acta*, 1797, 1477). Nevertheless, the involvement of the clusters FeS2-4 is not excluded.

Figure legend was modified accordingly and Asp169C was included in the scheme.

Comment 46

579-602. Full details needed for the reader to properly evaluate the work.

Response

Full details are now provided (lines 470-510).

Comment 47

632. *Details required.*

Response

Detailed methods are provided in the paper cited (Crooks *et al.* 2004 Genome Res. 14, 1188–1190).

Comment 48

666. *65 deg C is a very high temperature at which to study a protein. Indicate the functional stability of the protein under these conditions. How long is the protein held at this high temperature?*

Response

Rhodothermus marinus is a thermophilic organism with an optimal growth temperature of 65 °C. This is the reason we do our assays at this temperature. A sentence mentioning this is now included in the Methods section (line 578).

Comment 49

Fig. s2. Label subunits. Show density around hemes, FeS centers and coordinating residues.

Response

Supplementary Figure 2 was modified as suggested. Densities of a representative heme and iron-sulfur cluster are shown in panels *d* and *e*, respectively.

Comment 50

Fig. s3B. Unclear. Indicate that this is missing from the electron density and the complex structure reported.

Response

This figure was changed according to the reviewer's suggestions.

Comment 51

765. *Explain how E122 is stabilized in the membrane interior.*

Response

The figure legend was modified accordingly. Note that E122 is facing ActC and not exposed to the lipid environment.

Comment 52

Fig. s5C, s6, s7. Illegible.

Response

Supplementary Figure 5C is now shown separately in Supplementary Figure 7. Supplementary Figures 6 and 7 (now supplementary Figures 8 and 9) will be available at high resolution and should be properly visualized in the digital format.

Comment 53

Fig. s8. Report on the functional state of enzyme at the end of an ITC measurement at 65 C.

Response

As mentioned above, *Rhodothermus marinus* is a thermophilic organism with an optimal growth temperature of 65 °C. This is the reason we do our assays at this temperature. A sentence mentioning this is now included in the legend of the figure.

Comment 54

Table s1. List the 10 ligands.

Response

The ligands are now listed as suggested.

Reviewer #3

This manuscript presents a cryo-EM structure of Alternative Complex III, which is a relatively recently discovered electron transport complex found in many bacteria. It appears to be a functional replacement for the cyt bc1 complex (complex III), but structurally and evolutionarily unrelated.

The determination of a structure for this complex is exciting and I support publication of this landmark achievement. However, I do have some questions and suggestions that I think will

improve the manuscript. Most importantly is the question of whether or not ACIII is really a proton-pumping complex. I am not aware of any experiments that show this. The manuscript seems to assume that this is true, but the structure doesn't really give much support to that idea. The putative proton transfer channels are far removed from the electron transfer chains, and while that can certainly be the case, as in Complex I, evidence is needed that such proton transfer chains exist. As far as I can tell, the structure is entirely compatible with a mechanism in which the quinone is oxidized and the two protons ejected to the periplasm, but with no other protons pumped. This might be a more primitive type of complex and the bc1 complex with the Q cycle that pumps protons could be an improved version.

Response

We thank Reviewer #3 for his/her critical review of the manuscript. We are happy the Reviewer finds our work a "landmark achievement". We do agree the most important is the question of whether ACIII is really a proton-pumping complex. The structure here described is compatible with a redox-driven proton translocation mechanism, supported by the thermodynamics as discussed in the following comment.

Comment 1

Some discussion of the thermodynamics of the electron flow in the complex seems to be appropriate. The potential drop from the quinone to the terminal cytochrome acceptor seems to be relatively shallow, compared to either the bc1 complex or either Complex I or Complex IV. If proton pumping is going to take place, the energy to drive it ultimately has to come out of the redox gradient. Is there enough energy here to drive proton pumping?

Response

ACIII uses menaquinol as electron acceptor. The reduction potential of menaquinone is -70 mV (which compares to +120 mV for ubiquinone). The reduction potential determined for the HIPIP, the cytochrome *c* or the cytochrome *c* domain of the *caa*₃ oxygen reductase, i.e. of the electron acceptors are close to +250 mV (Pereira *et al* 1999 *Biochemistry* 38, 1276; Stelter *et al* 2008 *Biochemistry*. 47, 11953; Srinivasan *et al* 2005 *J. Mol. Biol.* 345, 1047). In this case the ΔE is 320 mV, since menaquinol gives two electrons, the energy available from the reaction is double this value. The membrane potential determined for *E. coli* is between 140 and 160 mV, depending on anaerobic or aerobic metabolism, respectively (Tran and Uden, 1998 *Eur J. Biochem.* 251, 538). Thus menaquinol:cyt *c* oxidoreduction can provide energy for pumping up

to four protons. A sentence referring to these thermodynamic considerations is now incorporated in the manuscript (lines 460 to 463).

Reviewers' comments:

Reviewer #1 (Remarks to the Author):

The manuscript has been revised adequately and can be accepted now.

Reviewer #2 (Remarks to the Author):

The authors have addressed many of the points raised. However, several items still require attention.

It would be useful to include a surface representation figure to supplement the video as requested by Reviewer #1.

The video shows side chains for D169 and D253. Are these in density? If not they should be identified in the text and legends as not in density and as having been modelled.

Legends for the videos should be included. These should explain what is being described in the various scenes, the labels used and most importantly what is based on reliable density and what has been interpreted.

Residues in the videos should be labelled for clarity including the arginines referred to by Reviewer #1.

Indicate if reliable density is or is not available for the side chain of E122 to make such a statement. Correct all text, other figures and videos likewise.

In Fig 7 there is an 8 Å line drawn between D169 and E122. If density is not available for either then the range of distances possible is considerable. Missing density for residues like these call into question helix rotation. These issues should be commented on in the ms.

Reviewer #2

Comment 1. Indicate if the protein was used fresh or frozen for EM work and the protein concentration used for negative staining.

Indicate where ActH might migrate in the SDS-PAGE in SFig 1. Is there a band on the gel associated with ActH?

State the identity of the very high and very low MW bands in the SDS-PAGE.

Given that *caa3* is modelled into the supercomplex its presence or absence in the SDS-PAGE should be commented on in the ms.

Comment on which of the bands in the SDS-PAGE were identified individually by mass spec.

Comment 11. This paper will be read by non-experts. It is important that they are informed as to what in the model is reliable and what is not. Accordingly, side chains and backbones identified as playing a functional or structural role that are not defined in density must be identified whenever they are referred to in the main text and in ALL figure and video legends. It is important not to give a false impression regarding the reliability of the model if it cannot be supported by convincing electron density. Perhaps model that is not in density could be shown in a way that is different from model that is in good density.

Comment 18. The figure would appear to be incomplete.

Comment 19. Information regarding radiation damage should be included in the ms.

Comment 26. Show docking even if it done manually. The purpose is to show there is the space and the appropriate chemistry in the proposed binding pocket for the quinol.

Add units for K_a .

Comment 29. Provide the requested information for the proposed model in Fig. 2c and include it in the ms.

Comment 31. The distance of transfer would appear to be quite large (Fig. 2c). Is it consistent with direct transfer? This issue should be addressed in the ms.

Comment 35. What else is in the sample? We know it contains ACIII and presumably *caa3* although this is not indicated in the SDS-PAGE in SFig 1. What else does it contain?

Were any of the other components considered as possible partners for supercomplex formation? This should be commented on in the ms.

Comment 39. This comment regarding glutamates and aspartates being rarely seen due to radiation damage must be included in the ms given the important functional role many play.

Comments 48 and 53. Comment in the ms regarding how long the protein remains at 65 C and its functional state at the end of the ITC measurement.

Comment 51. Explain how E122 is stabilized in the membrane.

We thank the Reviewers for their detailed comments that have helped us to improve the manuscript. Here is our detailed point-by-point response.

Reviewer #1

The manuscript has been revised adequately and can be accepted now.

Response

We thank Reviewer #1 for his/her critical review of the manuscript. We are happy the Reviewer finds “The manuscript has been revised adequately and can be accepted now”.

Reviewer #2

The authors have addressed many of the points raised. However, several items still require attention.

Response

We thank Reviewer #2 for his/her critical review of the manuscript.

Comment 1

It would be useful to include a surface representation figure to supplement the video as requested by Reviewer #1.

Response

This is now included as a Supplementary Figure 8.

Comment 2

The video shows side chains for D169 and D253. Are these in density? If not they should be identified in the text and legends as not in density and as having been modelled.

Response

The requested information is now included in the legend of Movie 2 (lines 828-829, Sousa_etal_revised_marked file).

Comment 3

Legends for the videos should be included. These should explain what is being described in the various scenes, the labels used and most importantly what is based on reliable density and what has been interpreted.

Response

Legends for the videos are now included as suggested (lines 813-814 and 818-830, Sousa_etal_revised_marked file).

Comment 4

Residues in the videos should be labelled for clarity including the arginines referred to by Reviewer #1.

Response

Residues shown in Movie 2 are now labeled.

Comment 5

Indicate if reliable density is or is not available for the side chain of E122 to make such a statement. Correct all text, other figures and videos likewise.

Response

We removed all text that suggests seeing density for a glutamate and aspartate side chain and added disclaimers where side chains are shown. Corrections are made in lines 96-101, line 245, lines 316-318, legends of figures 3, 5, 6 and 7 and legends of movie 2 (Sousa_etal_revised_marked file).

Comment 6

In Fig 7 there is an 8 Å line drawn between D169 and E122. If density is not available for either then the range of distances possible is considerable. Missing density for residues like these call into question helix rotation. These issues should be commented on in the ms.

Response

The line indicating the approximate distance between E122 and D169 has been removed from Figure 7. As we stated before, the distances are approximate. A comment regarding the

uncertainty of the modeling of the side chains of E122 and D169 is now included in line 316-318 (Sousa_etal_revised_marked file).

Absence of carboxylate side chains is a well-known issue in cryo-EM. It was first noted in high-resolution structures (Allegretti et al., eLife 3, e01963, 2014 and Bartesaghi et al., PNAS 111, 11709-11714, 2014) and since then in all cryo-EM structures. The issue was recently reviewed by us in Vonck and Mills, Curr. Opin. Struct. Biol. 46, 48-54, 2017. We have added these references to the manuscript (line 101, Sousa_etal_revised_marked file). However, although the absence of these side chains makes it impossible to model the side chain position accurately, it does not affect the chain tracing, as suggested by the reviewer. Helix rotation is fixed by other side chains and the backbone position is never in question.

Comment 7

Comment 1. Indicate if the protein was used fresh or frozen for EM work and the protein concentration used for negative staining.

Response

This is now indicated in lines 441 and 443, respectively (Sousa_etal_revised_marked file).

Comment 8

Indicate where ActH might migrate in the SDS-PAGE in SFig 1. Is there a band on the gel associated with ActH?

Response

ActH is expected to migrate at a MW of ~20 kDa. This is now indicated in the respective legend (line 9 in Supplementary Material, Sousa_etal_supplementary_revised_marked file).

Comment 9

State the identity of the very high and very low MW bands in the SDS-PAGE.

Response

The blue native gel and the SDS-PAGE are from the same sample, the one used for cryo-EM. All the bands present in the SDS-PAGE correspond to the complex shown in the blue native and to that for which we obtained the structure. The band at high MW results from an incomplete denaturation of the protein due to its stability (as stated in the manuscript, the optimal temperature of ACIII is 65 °C). The lowest “band” in the gel corresponds to the sample front.

This is now stated in the respective legend (lines 7-9 in Supplementary Material, Sousa_etal_supplementary_revised_marked file).

Comment 10

Given that caa3 is modelled into the supercomplex its presence or absence in the SDS-PAGE should be commented on in the ms.

Response

As we mentioned in the text (line 102, Sousa_etal_revised_marked file) and showed in Supplementary Fig. 1, the supercomplex accounts for only 7% of the particles. Given that caa₃ is smaller than ACIII, caa₃ would be only ~3 % of the protein sample and visible bands on the gel are not expected. This is now commented on in the legends of Supplementary Fig. 1a (lines 9-11 in Supplementary Material, Sousa_etal_supplementary_revised_marked file).

Comment 11

Comment on which of the bands in the SDS-PAGE were identified individually by mass spec.

Response

All the bands labeled in the gel were previously identified individually by mass spec and Edman degradation. This information is available in our previous works: Pereira et al, FEBS Letters 581, 4831, 2007 and Refojo et al., BBA 1797, 1477, 2010. This is now mentioned in the legend of Supplementary Fig. 1 (line 6-7 in Supplementary Material, Sousa_etal_supplementary_revised_marked file).

Comment 12

Comment 11. This paper will be read by non-experts. It is important that they are informed as to what in the model is reliable and what is not. Accordingly, side chains and backbones identified as playing a functional or structural role that are not defined in density must be identified whenever they are referred to in the main text and in ALL figure and video legends. It is important not to give a false impression regarding the reliability of the model if it cannot be supported by convincing electron density. Perhaps model that is not in density could be shown in a way that is different from model that is in good density.

Response

We have identified the side chains as stated in comment 5, most notably in lines 96-101 and figure legends (Sousa_etal_revised_marked file). All backbone tracings are always reliable.

Comment 13

Comment 18. The figure would appear to be incomplete.

Response

Supplementary Figure 5 was changed to show the whole structure of ACIII.

Comment 14

Comment 19. Information regarding radiation damage should be included in the ms.

Response

This information is now included in the manuscript (line 96-101, Sousa_etal_revised_marked file).

Comment 15

Comment 26. Show docking even if it done manually. The purpose is to show there is the space and the appropriate chemistry in the proposed binding pocket for the quinol.

Response

Considering the absence of density for carboxylate side chains, as mentioned in comments 2, 5, 6, 12, and 21, we consider docking to establish the chemistry too speculative. However, the homologous protein PsrC was shown to accommodate a quinone/quinol in the referred pocket by co-crystallization with menaquinone. We have added Supplementary Figure 7, showing an overlay of the two structures and comment on it in the manuscript on lines 252-255 (Sousa_etal_revised_marked file). Together, we consider this as evidence that there is enough space and the appropriate chemistry in the proposed binding pocket for the quinol from ActC.

Comment 16

Add units for K_a .

Response

The number refers to $\log K_a$ which is dimensionless.

Comment 17

Comment 29. Provide the requested information for the proposed model in Fig. 2c and include it in the ms.

Response

The requested information is "distances and orientation of electron donors and acceptors in the supercomplex. Describe the shape, charge complementarity between interacting surfaces." Considering the low resolution of the supercomplex (20 Å), such measurements depend on the docking of the two complexes and thus would be unreliable and speculative, so we choose not to include them.

Comment 18

Comment 31. The distance of transfer would appear to be quite large (Fig. 2c). Is it consistent with direct transfer? This issue should be addressed in the ms.

Response

The heme-heme distance in the fitting as shown is ~25 Å, but as said, it depends on the docking of two complexes in a low-resolution map (20 Å). This was addressed in the manuscript (lines 367-370, Sousa_etal_revised_marked file): "Even though the accuracy of fit in a 20 Å map does not allow the determination of minimum distances between the two hemes, their orientation is compatible with the direct reduction of *caa*₃ oxygen reductase by ActE, as we previously proposed (ref.20)". In the legends of figure 2c we state that "the relative position of ActE and subunit Ilc from the terminal oxidase in the supercomplex is favourable for direct electron transfer", but we do not comment on the distance. Furthermore, as we state in the introduction "ACIII can directly transfer electrons from quinol to the *caa*₃ terminal oxidase without the intervention of any soluble electron carrier (ref.8)" (lines 61-62, Sousa_etal_revised_marked file).

Comment 19

*Comment 35. What else is in the sample? We know it contains ACIII and presumably *caa*₃ although this is not indicated in the SDS-PAGE in SFig 1. What else does it contain?*

Response

In the mass spec analysis, we find dozens of *R. marinus* proteins, including several ATP synthase subunits, complex I subunits, outer membrane proteins, transporters, chaperonin,

etc. As we stated in our first letter, the detection limit of an optimized bottom-up LC-MS/MS experiment is today in the range of attomoles of a protein. This exceeds visualization on gels, and will lead to the identification of trace amounts of contaminating proteins. The presence of small amounts of contaminants does not affect the cryo-EM analysis, as evidenced by our successful reconstruction of a map that led to the structure determination of an unknown complex.

Comment 20

Were any of the other components considered as possible partners for supercomplex formation? This should be commented on in the ms.

Response

The other components were not considered, as we knew that ACIII and *caa*₃ form a supercomplex. This was stated in the introduction (lines 61-67) and we now added it to the results section (line 104-105) (*Sousa_etal_revised_marked* file). Fitting our model of ACIII and the crystal structure of *caa*₃ in the supercomplex map showed the proximities of subunits predicted by biochemical and functional studies (see fig. 7 of Refojo et al., BBA 1797, 1477-1482, 2010 below).

Fig. 7. Schematic representation of the structural and functional association between the alternative complex III (subunits A-G) and the *caa*₃ oxygen reductase (catalytic subunits I and II). The gray spheres represent c-type hemes, the smaller gray and black spheres represent copper ions while cubes and pyramids represent [4Fe-4S]^{2+/1+} and [3Fe-4S]^{1+/0} clusters, respectively.

Comment 21

Comment 39. This comment regarding glutamates and aspartates being rarely seen due to radiation damage must be included in the ms given the important functional role many play.

Response

The comment is now included as suggested (lines 96-101, *Sousa_etal_revised_marked* file).

Comment 22

Comments 48 and 53. Comment in the ms regarding how long the protein remains at 65 C and its functional state at the end of the ITC measurement.

Response

The protein activity is unchanged after incubation at 65 °C, during the ITC measurements and this is now stated in lines 534-535, *Sousa_etal_revised_marked* file. Protein activity is negligible at room temperature. The activity is maximum at 65 °C, in agreement with the optimal growth temperature of *Rhodothermus marinus*.

Comment 23

Comment 51. Explain how E122 is stabilized in the membrane.

Response

As mentioned in the text (line 314-316, *Sousa_etal_revised_marked* file) and shown in Figure 7, the highly conserved E122 is not exposed to the membrane, but is facing ActC and neighbors conserved residues from the other membrane subunits, including several serines, histidines and tyrosines, close to the putative quinol binding site. The environment of E122 presumably also contains water molecules, which are not visible at this resolution.

REVIEWERS' COMMENTS:

Reviewer #2 (Remarks to the Author):

The issues raised in my review have been addressed adequately in this revision.